# Neuronal Nsun2 deficiency produces tRNA epitranscriptomic alterations and proteomic shifts impacting synaptic signaling and behavior

J. Blaze[1,2], A. Navickas[3], H. L. Phillips[4], S. Heissel[5], A. Plaza-Jennings[6], S. Miglani [3], H. Asgharian[3], M. Foo[7], C. D. Katanski[7], C. P. Watkins[7], Z. T. Pennington[1,2], B. Javidfar[2,6], S. Espeso-Gil [2,6], B. Rostandy[5], H. Alwaseem[5], C. G. Hahn[8], H. Molina[5], D. J. Cai [1,2], T. Pan[7], W. D. Yao [4], H. Goodarzi [3], F. Haghighi[1,2,6,9] & S. Akbarian [1,2,6✉]

Epitranscriptomic mechanisms linking tRNA function and the brain proteome to cognition and complex behaviors are not well described. Here, we report bi-directional changes in depression-related behaviors after genetic disruption of neuronal tRNA cytosine methylation, including conditional ablation and transgene-derived overexpression of Nsun2 in the mouse prefrontal cortex (PFC). Neuronal Nsun2-deficiency was associated with a decrease in tRNA m$^5$C levels, resulting in deficits in expression of 70% of tRNA$^{Gly}$ isodecoders. Altogether, 1488/5820 proteins changed upon neuronal Nsun2-deficiency, in conjunction with glycine codon-specific defects in translational efficiencies. Loss of Gly-rich proteins critical for glutamatergic neurotransmission was associated with impaired synaptic signaling at PFC pyramidal neurons and defective contextual fear memory. Changes in the neuronal translatome were also associated with a 146% increase in glycine biosynthesis. These findings highlight the methylation sensitivity of glycinergic tRNAs in the adult PFC. Furthermore, they link synaptic plasticity and complex behaviors to epitranscriptomic modifications of cognate tRNAs and the proteomic homeostasis associated with specific amino acids.

[1] Department of Neuroscience, Icahn School of Medicine at Mt. Sinai, New York, NY, USA. [2] Friedman Brain Institute, Icahn School of Medicine at Mt. Sinai, New York, NY, USA. [3] Department of Biochemistry and Biophysics, University of California San Francisco, San Francisco, CA, USA. [4] Departments of Psychiatry and Behavioral Sciences, Neuroscience and Physiology, Upstate Medical University, Syracuse, NY, USA. [5] The Rockefeller University Proteomics Resource Center, The Rockefeller University, New York, NY, USA. [6] Department of Psychiatry, Icahn School of Medicine at Mt. Sinai, New York, NY, USA. [7] Department of Biochemistry and Molecular Biology, University of Chicago, Chicago, IL, USA. [8] Department of Neurosciences, Thomas Jefferson University, Philadelphia, PA, USA. [9] Research and Development Service, James J. Peters Veterans Affairs Medical Center, Bronx, NY, USA. ✉email: schahram.akbarian@mssm.edu

Targeting the brain's translational machinery bears promise for psychiatric disease treatment[1–3] but the role of tRNAs —76–90 nucleotide cloverleaf-shaped structures and key players for ribosomal protein synthesis comprising ~10% of the cell's RNA pool—remains unexplored. This is surprising given that neurological phenotypes are often the primary manifestation of mutations affecting the tRNA regulome[4–6]. For example, mutations in a subset of mitochondrial enzymes charging tRNAs with their cognate amino acids have been associated with adult-onset frontal lobe dysfunction, depression, and cognitive decline[7,8]. Furthermore, a recent metabolomics meta-analysis integrating findings from multiple mouse and rat stress-based depression models listed tRNA charging and amino acid metabolism among the top five ranking pathways significantly affected in the depressed brain, together with endocannabinoid signaling, catecholamine biosynthesis, and GABA receptor signaling[9]. Here, we focus on adult brain phenotypes after genetic disruption of tRNA epitranscriptomic modification by NOL1/NOP2/SUN domain tRNA cytosine methyltransferase (Nsun2), which is essential for cytosine methylation ($m^5C$) at the tRNA's variable loop on >75% of actively transcribed mammalian tRNAs[10].

Evidence for the importance of solely NSUN2-mediated $m^5C$ in human health and disease was first provided by individuals with a loss-of-function mutation in *NSUN2*, displaying intellectual disability (ID), facial dysmorphism, and distal myopathy[11–14]. The idea that Nsun2 deficiency causes neurological abnormalities was further delineated by studies in *Drosophila* (d) and mouse[10,15], including knockdown of *Drosophila* Nsun2 (dNsun2) which impaired short-term memory after aversive olfactory conditioning, and the phenotype was rescued by pan-neuronal expression of dNsun2[11]. Furthermore, mice with *Nsun2* germline deletion demonstrated various impairments in locomotor activity and behavior together with reduced brain size due to excessive cell death in the prenatal brain[10]. The mechanism thus far suggested for these deficits is impaired translation induced by increased tRNA fragmentation after Nsun2 ablation[10,16]. While there is evidence for global impairment of protein synthesis via fluorescent labeling of nascent proteins in embryonic mouse brain[10] and adult mouse skin[10], to our knowledge no one has used unbiased proteomic approaches such as mass spectrometry to directly measure specific molecular pathways altered by Nsun2 ablation in any tissue type. It has been established previously that global Nsun2 knockout in mouse alters the pool of mature tRNAs in the embryonic brain[10] and adult skin[10] and liver[17], including a marked depletion of tRNA$^{Gly}$, which is a key target of Nsun2 methylation. However, although tRNA cytosine methylation ($m^5C$) has recently been profiled in various tissues, including embryonic brain[10,16–25], the molecular characterization of the Nsun2-dependent epitranscriptome, and its functional relevance in the mature adult mammalian brain is still completely unexplored. In this study, we use three different genetic approaches selectively targeting Nsun2 function in differentiated neurons of the postnatal and adult brain. These include neuron-specific conditional *Nsun2* ablation and transgene-mediated increase in *Nsun2* expression and methylation activity in the adult prefrontal cortex (PFC), thereby focusing on the effect of Nsun2 enzymatic activity on specific neuronal subpopulations in the mature brain. We report that Nsun2 shapes complex behaviors and neuronal function, which are highly sensitive to bi-directional changes in tRNA methyltransferase activity. We show that the underlying mechanisms include alterations in tRNAs defined by high cytosine methylation specifically in the variable loop region, resulting in prominent deficits of tRNA$^{Gly}$ isodecoders with corresponding shifts in the neuronal proteome due to decreased translational efficiency of glycine-rich neuronal proteins. Ultimately, these distortions in the glycinergic

neuronal translatome lead to a 2.46-fold increase in PFC glycine levels associated with multifold increases in several key enzymes of the glycine biosynthetic pathway, further illustrating that in the adult brain, proteomic and metabolic homeostasis associated with specific amino acids is linked to epitranscriptomic modification of cognate tRNAs.

## Results

**Neuronal Nsun2 modulates cortical tRNA cytosine methylation.** We generated mice with conditional neuron-specific ablation of Nsun2 in the forebrain (*CamK-Cre$^+$,Nsun2$^{2lox/2lox}$* mutant mice), (Fig. 1a and Supplementary Fig. 1a–c). Nsun2 mutant mice were born and survived into adulthood at expected Mendelian ratios (Supplementary Fig. 1d) without overt health abnormalities and showed ~10% reductions in body and brain weight compared to WT littermates 14-16 weeks after birth (ANOVA; body weight, $P < 0.001$; brain weight, $P < 0.001$), with female but not male mutants exhibiting decreased brain/body ratio ($t$ test; $P < 0.01$; Supplementary Fig. 1d and Supplementary Table 1). Assessment of Nsun2 protein levels in mutant vs. control using western blot confirmed a significant decrease in Nsun2 protein in tissue homogenate. Nsun2 was still detectable at moderate levels in homogenate due to expression in other, non-*Camk2a*-expressing cell types (inhibitory neurons, glia, etc.), as our knockout was specific to excitatory neurons expressing *Camk2a*, which encompasses the majority of cortical neurons. tRNA bisulfite sequencing (Supplementary Fig. 2a–c) for five highly expressed tRNAs in the adult cerebral cortex (out of 162 detected cytosolic isodecoders (tRNA molecules sharing the same anticodon but differ in sequence elsewhere); see Fig. 2a for tRNA expression levels independent of epitranscriptomic modification), including tRNA$^{Gly}_{GCC}$, tRNA$^{Glu}_{TTC}$, tRNA$^{Glu}_{CTC}$, tRNA$^{Asp}_{GTC}$, tRNA$^{Val}_{AAC}$, and tRNA$^{Pro}_{TTG}$, which all consistently showed >50% methylation deficits at sites of Nsun2-sensitive cytosine(C) residues, including C46-49 in *CamK-Cre$^+$,Nsun2$^{2lox/2lox}$* mutant compared to littermate control mice (Fig. 1c, d, Supplementary Fig. 3a and Supplementary Table 2). To assess the specificity of these findings, we delivered into PFC of adult C57B6/J mice an adeno-associated virus vector 8 (AAV8)-transgenic expression cassette for chimeric *Nsun2-green fluorescent protein (Nsun2-GFP)* under control of a neuron-specific (*hSyn1*) promoter. Indeed, in *AAV8$^{hSyn1-Nsun2GFP}$* injected PFC specimens, only one cytosine residue with moderate baseline $m^5C$ levels (~10–20%), tRNA$^{Gly}_{GCC}$ C39, displayed a significant methylation increase ($t$ test, adjusted $P = 0.038$) (Fig. 1c, d and Supplementary Table 2). These opposing tRNA $m^5C$ changes after neuron-specific Nsun2 ablation and transgenic expression models were extremely specific, because non-Nsun2 regulated $m^5C$ residues, including the $m^5C$ Dnmt2-methylated tRNA$^{Gly}_{GCC}$ C37 and tRNA$^{Asp}_{GTC}$ C38[26,27] only showed minimal differences compared to controls (Fig. 1c, d and Supplementary Table 2). Furthermore, non-$m^5C$ tRNA methylation including $m^1A$, $m^3C$, and $m^1G$ modification, and levels of aminoacylation/charging for 151 nuclear-encoded and 21 mitochondrial tRNA isodecoders with detectable levels of expression in adult cortex revealed no significant differences between Nsun2-deficient and control cortex (Supplementary Fig. 4c, d).

**Nsun2 knockout selectively depletes tRNA$^{Gly}$ expression.** It has been previously established that tRNA $m^5C$ within the variable loop region, the primary Nsun2 target (Fig. 1c, d and ref. [28]), could affect tRNA stability and translation efficiency[29,30]. To explore, we profiled full-length tRNAs in adult *CamK-Cre$^+$, Nsun2$^{2lox/2lox}$* and control cortex with dm-HydrotRNA-seq[31,32]. 10/10 tRNA$^{Gly}$ isodecoders showed decreased expression in the

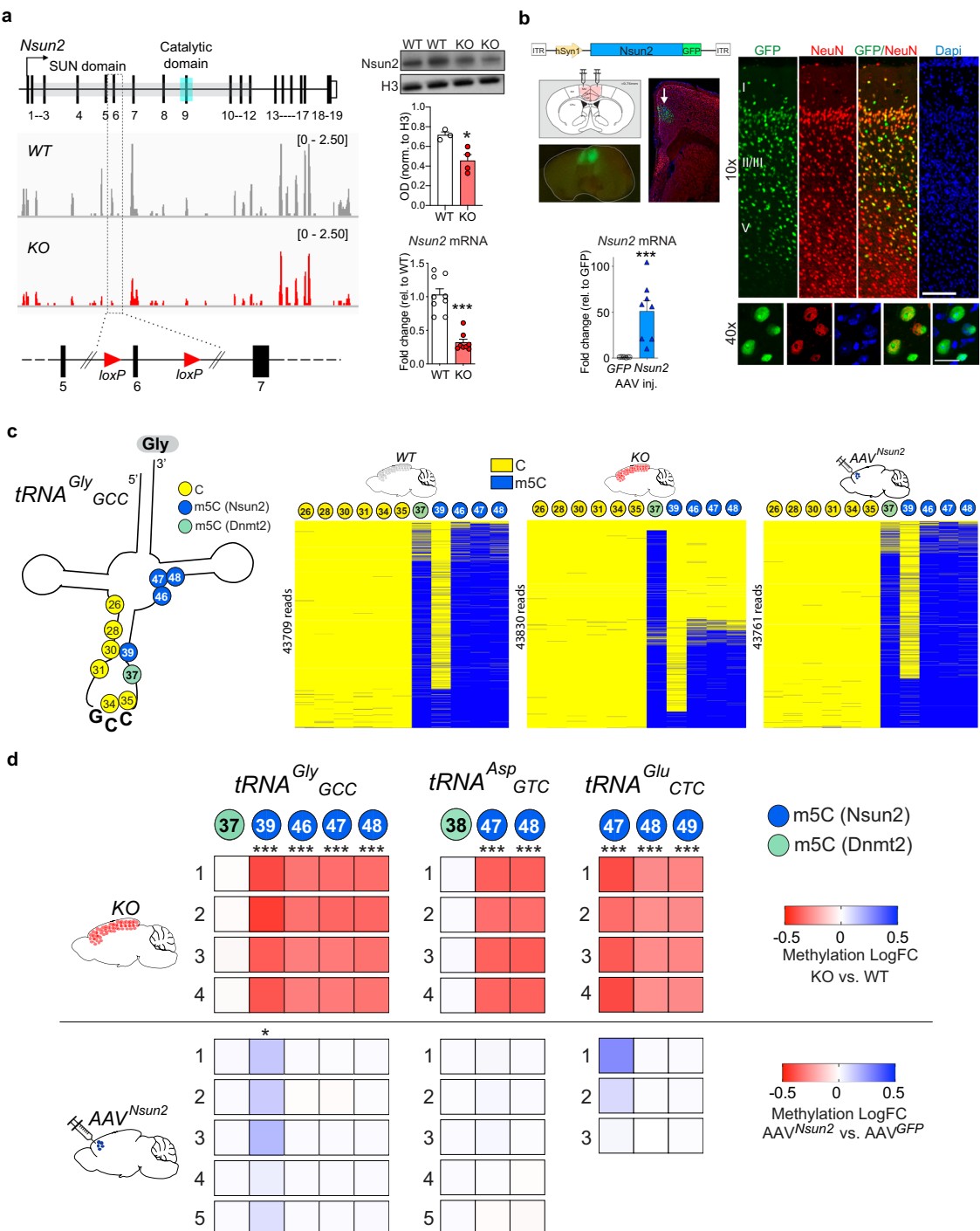

Nsun2-deficient cortex compared to control. This included significant deficits (FDR adj. $P < 0.05$) in 4/4 tRNA$^{Gly}_{GCC}$, 2/4 tRNA$^{Gly}_{CCC}$, and 1/1 tRNA$^{Gly}_{TCC}$ isodecoders. These changes were tRNA$^{Gly}$-specific, because only 4/152 (non-tRNA$^{Gly}$) isodecoders showed a significant change in the mutant, without anticodon preference (Fig. 2a, statistical analyses for all detected isodecoders in Source Data file). Next, we confirmed the highly specific deficits in tRNA$^{Gly}_{GCC}$, tRNA$^{Gly}_{CCC}$, and tRNA$^{Gly}_{TCC}$ isodecoder expression in our Nsun2-deficient cortex (FDR adj. $P < 0.05$) with an independent tRNA profiling method, YAMAT-seq[33] (Supplementary Fig. 4a). We then performed tRNA fragment (tRF)/tRNA half (tiRNA) sequencing (tRF 3'—17–22 nt; 5'—14–32 nt; and tiRNAs—33–35 nt), representing a

heterogenous small RNA group produced from precursor or mature tRNAs and important for translational regulation and cellular stress responses[34] and observed in Nsun2-deficient cortex a selective deficit in tRNA$^{Gly}_{CCC}$, tRNA$^{Gly}_{GCC}$, and tRNA$^{Gly}_{TCC}$ 5' originating fragments primarily representing 5' tiRNAs and 5'tRF-5c (28–32 nt long) ($P < 0.05$; Supplementary Fig. 4b). Therefore, our three independent sequencing assays exploring the "tRNAome", further confirmed by isoacceptor-specific qPCR (Supplementary Fig. 4a, b), consistently showed that loss of neuronal Nsun2 is associated, in a codon-specific manner, with selective deficits in multiple tRNA$^{Gly}$ isodecoders. In contrast, non-tRNA$^{Gly}$ isodecoders were minimally affected. Our findings resonate with previous studies in Nsun2- and Dnmt2-RNA

**Fig. 1 Neuronal Nsun2 expression and activity modulates tRNA cytosine methylation in the adult cortex. a** Left, schematic representation of conditional Nsun2 ablation with *loxP* sites surrounding *Nsun2* exon 6 and representative RNA-seq profiles from the cortex of adult *CamK-Cre+,Nsun2^2lox/2lox* (KO) and WT control mice. Right, immunoblot (top) for Nsun2 protein expression in cortical tissue homogenate (Histone H3 was used as a loading control; two-tailed, $t(5) = 3.381$, *$P = 0.0196$; $n = 3$ WT, 4 KO). Uncropped western blots depicted in Supplementary Fig 1b. *Nsun2* mRNA levels (bottom) were measured in forebrain tissue homogenate using qPCR with *Gapdh* as a housekeeping gene ($n = 9$ WT, 8 KO; two-tailed, Mann–Whitney $U = 0$, ***$P < 0.0001$). Data are presented as mean values $+/-$ SEM. **b** Top left, schematic representation of viral vector containing *hSyn1* promoter to drive neuronal expression, mouse *Nsun2* transcript, and *GFP* fusion protein (top). Brain atlas depiction of discrete PFC bilateral microinjections that were confirmed via GFP illumination (left) and immunostaining of Nsun2GFP (right; top scale bar = 100 μm, bottom scale bar = 20 μm). Note nucleolar staining pattern of Nsun2GFP. Bottom left, a significant increase in *Nsun2* mRNA was detected by qPCR in PFC tissue after *AAV8^hSyn1-Nsun2GFP* injection compared to *AAV8^hSyn1-GFP* controls using *Gapdh* as a housekeeping gene ($n = 7$–8/condition; two-tailed, Mann–Whitney $U = 0$, ***$P = 0.0003$). Data are presented as mean values $+/-$ SEM. **c** Left, schematic of tRNA^Gly_GCC, depicting 11 cytosines that were queried with tRNA bisulfite sequencing. Yellow circles represent unmethylated cytosines, blue circles represent cytosines methylated by Nsun2, and green circles represent cytosines methylated by Dnmt2. Right, representative tRNA bisulfite sequencing methylation maps for WT & Nsun2 KO cortex and *AAV^Nsun2* PFC. tRNA bisulfite methylation maps include one cytosine per column and reads are represented as horizontal lines. **d** Methylation maps for differential methylation levels of Nsun2 and Dnmt2 cytosine targets compared to appropriate control for tRNA^Gly_GCC, tRNA^Glu_CTC, and tRNA^Asp_GTC ($n = 3$–5 biological replicates per condition (denoted as 1–5), two-tailed $t$ test, ***FDR adj. $P < 0.001$; *FDR adj. $P = 0.038$ vs. same-batch control for each group; statistics for each cytosine shown in Supplementary Table 2). Source data are available as a Source Data file.

cytosine methyltransferase-deficient fibroblasts which showed isodecoder destabilization limited to a small subset of tRNAs including tRNA^Gly_GCC[35]. However, in contrast to our findings in the adult cortex, Nsun2 in the prenatal brain is essential for inhibition of tRNA fragment production[10].

**Nsun2 ablation decreases the expression of Gly-rich proteins.** Studies in dividing cells and tissues lacking Nsun2 expression reported broadly impaired protein synthesis[10,25,36]. Therefore, we explored with LC-MS/MS mass spectrometry the adult PFC proteome after neuronal Nsun2 ablation. From a total of 5820 proteins identified in the PFC, 635 were decreased and 853 increased in *CamK-Cre+, Nsun2^2lox/2lox* compared to littermate control PFC ($n = 5$–6/group; FDR adj. $P < 0.05$; Fig. 2b, Supplementary Fig. 4g). Strikingly, neuronal signaling and synaptic functioning pathways represented 10/10 top-ranking StringDB functional categories for the group of downregulated proteins, while pathways associated with RNA processing, metabolism, and tRNA aminoacylation/ligation ranked top among the list of upregulated proteins (Fig. 2c). Given the loss of codon-specific tRNA^Gly isodecoders in our Nsun2-deficient cortex, we then examined glycine amino acid composition in the cortical proteomes. Indeed, downregulated proteins were significantly glycine-enriched (average 7.16% glycine content) in the mutant compared to wild-type proteome, which on average had 6.63% glycine content ($P < 0.0001$; Fig. 2c). This included many proteins crucial for synaptic functioning, such as neurogranin (Nrgn), calcium voltage-gated channel auxiliary subunit gamma 8 (Cacng8), synaptogyrin 3 (Syngr3), neuronal pentraxin receptor (Nptxr), discs large MAGUK scaffold protein 4 (Dlg4), glutamate ionotropic receptor AMPA type subunit 2 (Gria2), and protein kinase C gamma (Prkcg) (comprehensive list in Supplementary Table 3). Conversely, proteins upregulated after Nsun2 ablation were significantly glycine de-enriched ($P = 0.046$; Fig. 2c). These alterations were highly specific because glycine content of the entire set of $n = 434$ neuron-specific proteins (UniProtKB) in our cortical proteome (comprised of neuronal and glial proteins) was very similar to the glycine content of the total proteome in our cortex homogenates (Mann–Whitney test, $P = 0.412$; Supplementary Fig. 4f), confirming that the observed enrichment of high glycine content proteins in the fraction of downregulated proteins in the Nsun2 mutant cortex do not reflect a non-specific decline of the neuronal proteome overall. Notably, a subset of 50 downregulated proteins known to be involved in excitatory synaptic transmission (Supplementary Table 3), including Neurogranin (Nrgn) as a key regulator of glutamate NMDA receptor-

dependent intracellular signaling in PFC[37,38], and calcium voltage-gated channel auxiliary subunit gamma 8 (Cacng8), regulating AMPA receptor intramembranous distribution[39], displayed even higher average glycine content (7.62%; Supplementary Fig. 4f). Therefore, deficits in neurotransmission due to deficiencies in Gly-rich synaptic proteins may be present in the Nsun2 mutant cortex.

Correlation analysis between proteome and transcriptome (RNAseq; $n = 2$/genotype) of the 76 synapse-associated genes comprising the top 10 GO categories (Fig. 2c and Supplementary Fig. 4e) demonstrated poor correlation ($r^2 = 0.1066$), suggesting that the aforementioned proteomic deficit in Nsun2-deficient cortex cannot be explained solely by gene expression alterations. We then computed codon-specific translational efficiency ratios (logTER) from normalized differentials of protein and RNA levels[40]. Indeed, we again identified a significant association between glycine codon content (including all four glycine codons due to their unanimous decrease in tRNA expression in the Nsun2 KO) and translation such that low logTERs were associated with higher glycine codon content within genes and vice versa (Fig. 2d). Interestingly, glutamic acid (Glu) and aspartic acid (Asp), two amino acids that in our set of 5820 cortical proteins were anti-correlated with Gly content, showed significantly increased logTERs in the mutant cortex (Supplementary Fig. 5a). Importantly, tRNA abundance is a key determinant for translational elongation rates during protein biogenesis, including codon-specific variabilities in ribosome speed along mRNAs[41–43]. Therefore, we asked whether the observed deficit in *Gly* tRNAs was associated with codon-specific alterations in ribosome dynamics. To examine this, we conducted genome-wide ribosomal footprinting assays (Riboseq)[44], calculated P-site offsets (the site holding the tRNA associated with the growing polypeptide chain) with riboWaltz[45], and then computed both codon-specific dwell times as a measure of the codon-specific decoding rate and the expression of codons occupying the A-site for WT vs. KO cortex (see "Methods"; Fig. 2e; Supplementary Fig. 5c, d). Indeed, the cerebral cortex of adult *CamK-Cre+, Nsun2^2lox/2lox* mice, in comparison to cortex with wild-type levels of Nsun2, showed a significant increase in dwell time (DT) ($P = 1.97e-71$) and A-site occupancy ($P = 3.07e-21$) for Gly (GGN) codons. This alteration was Gly-specific because codons from the 19 non-Gly amino acids were much less affected (Fig. 2f). Closer inspection of the 64 codons revealed that the excess DT and increased A-site occupancy in ribosomes from Nsun2 mutant cortex were primarily driven by the Gly GGA codon (TCC anticodon). Likewise, in the KO cortex A-site occupancy was also significantly increased for 2/3 of the

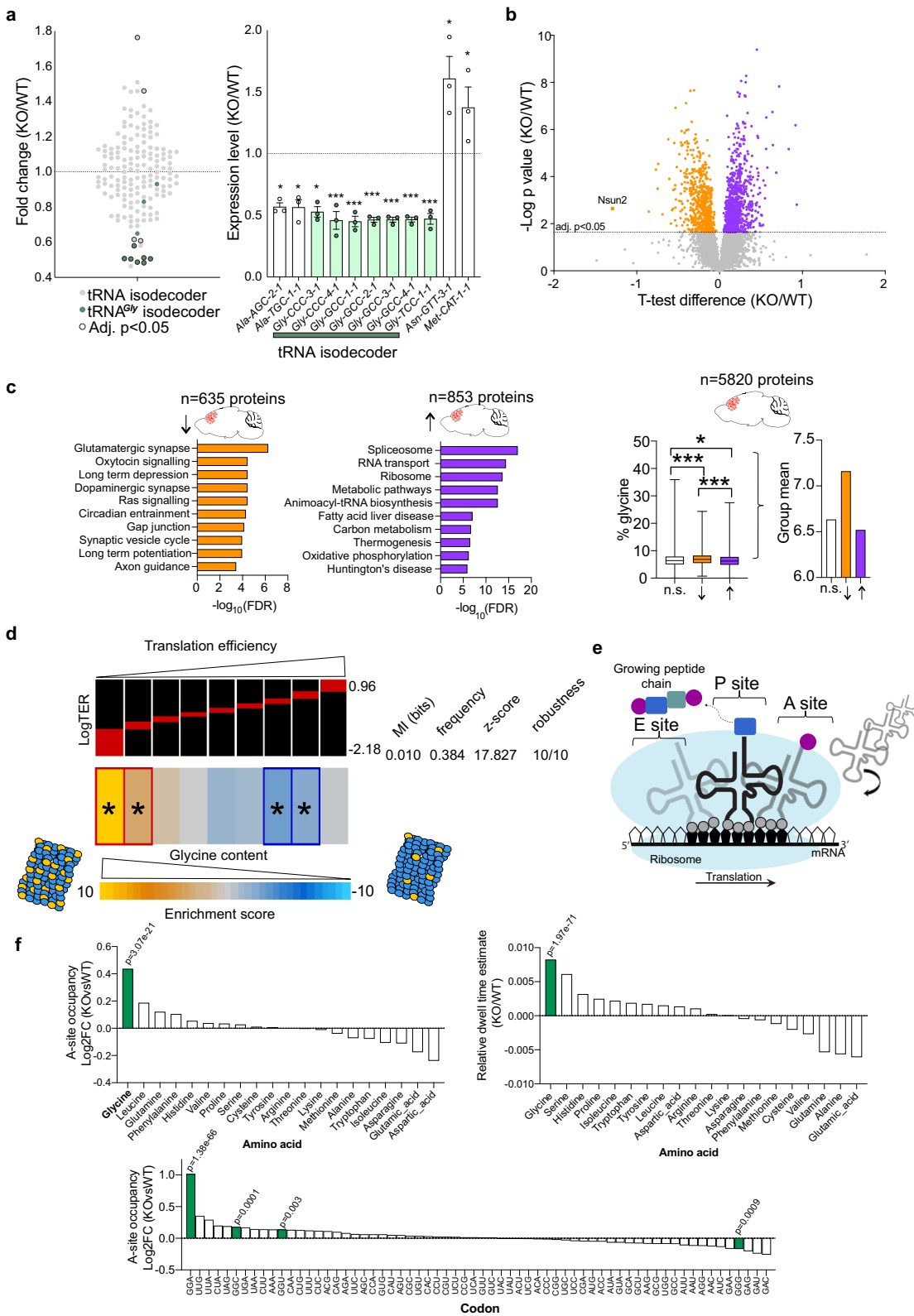

other Gly codons (GGC and GGU) (Fig. 2f and Supplementary Fig. 5e). Furthermore, GGA codon-enriched transcripts had lower translational efficiencies in KO as compared to the mutant cortex (Supplementary Fig. 5b). Finally, we examined ribosomal dynamics for the AAC and ATG codons associated with the two non-*Gly* tRNAs that were upregulated in the Nsun2 KO cortex (see Fig. 2a;

tRNA$^{Asn}_{GTT}$ and tRNA$^{Met}_{CAT}$). There were no significant changes in ribosome DT or A-site occupancy (Fig. 2f and Supplementary Fig. 5e). Therefore, we conclude that loss of glycinergic tRNAs, including tRNA$^{Gly}_{TCC}$ isodecoders, is associated with disruption, or slowing, of proper translational elongation at the site of the corresponding Gly codons.

**Fig. 2 Nsun2 ablation depletes tRNA$^{Gly}$ causing decreased expression of glycine-rich synaptic proteins. a** Left, tRNA sequencing of mouse forebrain in Nsun2 KO and WT ($n = 3$/genotype). Note decreased expression in 10/10 tRNA$^{Gly}$ isodecoders (green) among 162 total isodecoders (gray). Isodecoders that are differentially expressed (adj. $P < 0.05$) between KO and WT have black outlined circles. Right, 11 isodecoders were significantly altered by Nsun2 ablation (FDR $P < 0.05$) with 7/9 downregulated isodecoders belonging to the tRNA$^{Gly}$ family. Data are presented as mean values $+/-$ SEM. **b** Unbiased proteomic screen identified 1488 proteins significantly altered (FDR adj. $P < 0.05$) after Nsun2 ablation (635 decreased, 853 increased; $n = 5$–6/genotype). **c** Left, gene ontology analysis using StringDB identified the top ten significant KEGG pathways downregulated in Nsun2 KO including those involved in synaptic functioning and neurotransmission, while one of the top ten significant KEGG pathways upregulated was the family of aminoacyl tRNA ligases. Right, box plot for glycine content in three subgroups of proteins, n.s (not significant; 4332 proteins; median: 6.403, IQR: 2.811, whiskers represent min (0) and max (35.93)), decreased (635 proteins; median: 6.849, IQR: 2.7, whiskers represent min (0.73) and max (24.36)), and increased (853 proteins; median: 6.234, IQR: 2.852, whiskers represent min (0) and max (27.54)) proteins in Nsun2 KO cortex vs. WT. (two-tailed, Mann–Whitney test; \*\*\*$P < 0.0001$, \*$P = 0.046$). **d** Proteins produced by genes with higher glycine codon content exhibited a significant decrease after Nsun2 ablation. Left, LogTER (top, red) corresponding to glycine codon content enrichment scores (bottom, yellow-blue). Heatmap represents gene clusters from the top 50% of glycine codon-enriched genes, with yellow representing most enriched and blue representing least enriched. The red and dark-blue borders indicate the statistical significance of the calculated hypergeometric $P$ values for that cluster after Bonferroni's correction, obtained from the $z$ scores associated with the mutual information (MI) values calculated with 10,000 randomization tests (adj. $P < 0.05$). **e** Schematic of translation at the ribosome and locations of the A, P, and E sites. **f** Top, RiboSeq data for A-site occupancy (KO vs. WT) (top left) and relative ribosomal dwell time (KO/WT) (top right) grouped by amino acid. Bottom, ribosome A-site occupancy shown for all 64 codons. Note the significant increase in DT and A-site occupancy for Gly (GGN) codons and specifically GGA codons ($n = 2$ KO/2 WT). Source data are available as a Source Data file.

**Glycine biosynthesis is increased in the Nsun2 mutant cortex.** We noticed five differentially expressed enzymes in our proteomics screen that were involved in the monosaccharide-based glycine/serine biosynthesis and metabolism pathway, including phosphoserine aminotransferase 1 (Psat1), phosphoserine phosphatase (Psph), phosphoglycerate dehydrogenase (Phgdh), serine hydroxymethyltransferase 2 (Shmt2), and serine racemase (Srr) (Fig. 3a and Supplementary Table 4). Because all five glycine/serine biosynthesis proteins showed significantly altered expression in our mutant cortex in a direction consistent with increased glycine biosynthesis, we speculated that alterations in expression of these proteins combined with the impact of tRNA$^{Gly}$ dysfunction in the Nsun2 mutant cortex could signal changes in bulk amino acid content, specifically for glycine. To examine this, we used an unbiased metabolomic approach to profile 19 of the 20 canonical amino acids (no data for cysteine due to high reactivity). Indeed, there was a significant 2.46-fold (146%) increase of glycine in the Nsun2-deficient as compared to wild-type littermate cortex ($n = 4$/group, $t$ test $P < 0.05$). In contrast, levels for each of the remaining 18 amino acids showed minimal, non-significant differences between mutant and control cortex (Fig. 3b). These findings, taken together, point to selective susceptibility of the glycinergic translatome in the adult cerebral cortex upon removal of neuronal Nsun2. Nsun2 ablation resulted in decreased levels of hypomethylated Gly tRNAs, with decreased translation of Gly-rich proteins and compensatory upregulation of the glycine biosynthetic pathway, ultimately leading to a 2.46-fold increase in glycine levels in bulk tissue extracts.

**Neuronal Nsun2 deletion impacts synaptic transmission in PFC.** Of note, as discussed above, alterations in the cortical proteomic landscape after neuronal Nsun2 ablation included many Gly-rich proteins crucially important for synaptic signaling (Supplementary Table 3 and Supplementary Fig. 4f). Therefore, in order to test whether neurotransmission is affected in the Nsun2 mutant cortex, we performed whole-cell patch-clamp recordings on individual layer V pyramidal neurons in the mouse PFC to measure AMPA receptor-mediated miniature(m) excitatory postsynaptic currents (EPSCs) in adult Nsun2 mutants and controls. We specifically chose two sub-regions of the rodent PFC, the anterior cingulate cortex (ACC) and prelimbic cortex (PrL), which have been repeatedly implicated in rodent cognition[46–49] and depressive-like behavior[50,51]. In ACC, there was no change in amplitude ($t(23) = 0.9987$, $P = 0.328$) but a significant decrease in mEPSC frequencies ($t(23) = 2.958$, $P =$

0.007), indicating potential loss of functional synapses or a lower probability of presynaptic glutamate release in KO mice. In PrL, mEPSC frequencies were normal ($t(23) = 0.6787$, $P = 0.504$) but there was a significant decrease in amplitude for KO mice ($t(23) = 2.243$, $P = 0.035$), suggesting a decreased abundance of postsynaptic AMPA receptors or less neurotransmitter quanta in presynaptic vesicles (Fig. 4a). To more directly link functional changes in synaptic transmission with alterations in Gly-rich synaptic proteins, we isolated synaptosomes from Nsun2 KO cortex and measured protein abundance for the synaptic protein with the highest glycine content, Neurogranin (Nrgn; Supplementary Table 3, Fig. 4b, and Supplementary Fig. 5f). Synaptic fractions in the KO cortex indeed had significantly decreased Neurogranin protein expression ($t(8) = 2.282$, $P = 0.026$, one-tailed; Fig. 4b). Based on these results, we propose a working model linking neuronal codon-specific loss of Gly tRNAs after Nsun2 ablation to impaired translation of synaptic proteins, resulting in defective synaptic plasticity and signaling (working model, Fig. 4c).

**Neuronal Nsun2 deletion impairs complex behaviors.** Given these alterations in PFC neurotransmission, we asked whether PFC-dependent behaviors could be affected in mice with neuronal Nsun2-deficiency. To this end, we subjected multiple cohorts of mice to behavioral realms causally related to cortical function, including fear learning, anxiety, and behavioral despair/depression[50,52–54] (Supplementary Table 5). Indeed, $CamK\text{-}Cre^+$, $Nsun2^{2lox/2lox}$ mice showed impaired acquisition of a tone/shock association in the fear-conditioning paradigm ($P = 0.06$) and a dramatic reduction in freezing when placed in the training context during the contextual fear memory test ($P < 0.001$) suggesting impaired contextual memory (Fig. 5a). This cognitive deficit was specific to contextual memory because cued fear conditioning and Y-maze working memory were completely preserved (Fig. 5a). However, mutant mice were significantly less anxious in the elevated plus maze ($P = 0.004$) and light/dark box ($P = 0.012$; Fig. 5b and Supplementary Fig. 6a–c) compared to controls and exhibited significantly decreased immobility scores on the forced swim test ($P = 0.009$) but not the tail suspension test ($P = 0.291$; Fig. 5c).

**Nsun2 overexpression in PFC produces depressive-like behavior.** We were highly surprised to observe reductions in depression and anxiety-related behaviors in mice with neuronal Nsun2-

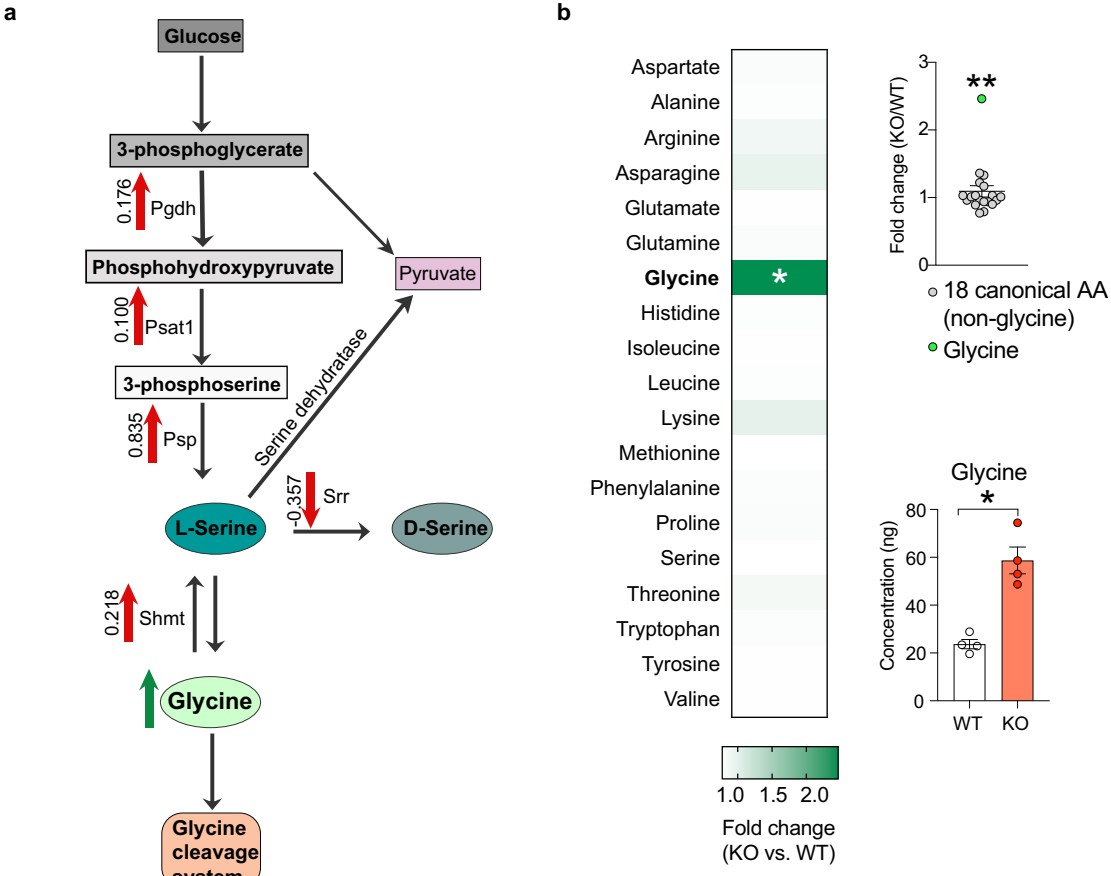

**Fig. 3 Nsun2 ablation alters glycine–serine biosynthesis pathways leading to increased glycine amino acid concentrations in the mutant cortex. a** Schematic of glycine–serine biosynthesis pathways that are significantly altered in cortex after Nsun2 ablation. Red arrows represent significant FDR adjusted $P$ values <0.1 in unbiased proteomic screen (see Fig. 2b) between Nsun2 KO and WT cortex with average $t$ test differences denoted next to arrows. The green arrow represents significantly changed amino acid concentration in KO vs. WT cortex. **b** Metabolomic profiling of mutant and WT cortex for 19 amino acids (20 canonical amino acids, except for cysteine; see "Methods") revealed a significant increase in glycine for Nsun2 KO vs. WT ($n = 4$/ genotype; two-tailed $t$ tests, FDR adj. *$P = 0.020$; **Grubb's test for fold changes identified glycine as a significant outlier with alpha = 0.001). Data are presented as mean values $+/-$ SEM. Source data are available as a Source Data file.

deficiency, and we then wondered whether a viral-mediated increase in neuronal Nsun2 expression in the adult cortex would elicit similar or opposite phenotypes. Indeed, mice expressing a *Nsun2GFP* transgene ($AAV8^{hSyn1-Nsun2GFP}$; Fig. 1b) bilaterally in PFC neurons showed significantly increased immobility scores in the two behavioral despair tests ($P = 0.005$ and $0.003$; Fig. 5d), while anxiety-related test scores were indistinguishable from controls (Supplementary Fig. 6a–c). In addition, fear acquisition, contextual memory, cued fear memory, and Y-maze working memory were all preserved in Nsun2 overexpressing mice (Supplementary Fig. 6d).

**Nsun2 ablation in PFC produces an anti-depressant phenotype**. Due to these bi-directional behavioral effects, with antidepressant-like phenotypes after Nsun2 ablation in postnatal forebrain but increased depression-like behaviors in mice expressing a *Nsun2* transgene selectively in adult PFC neurons, we next sought to determine if we could recapitulate the anti-depressant-like effect with a PFC-specific neuronal Nsun2 ablation in adulthood. To this end, we bilaterally injected $AAV^{hSyn1-CreGFP}$ into PFC of 10–12 weeks old adult $Nsun2^{2lox/2lox}$ mice (Supplementary Fig. 1e). PFC-specific loss of *Nsun2* expression was confirmed by qPCR ($n = 16$–17/AAV injection type; $t(31)$:

3.945, $P < 0.001$) and the resulting loss of Nsun2 methyl-transferase activity further confirmed by tRNA bisulfite sequencing for $tRNA^{Gly}_{GCC}$, showing decreased methylation at Nsun2 target cytosines ($n = 3$/AAV injection type; C39: $P = 0.032$, C46,47,48: $P = 0.002$; Supplementary Table 6). Indeed, mice with PFC-specific loss of Nsun2 showed decreased immobility in tests of behavioral despair, specifically the TST ($N = 7$–10/sex/AAV injection type; $t$ test, $P < 0.001$; Fig. 5e and Supplementary Table 5).

## Discussion

We show that Nsun2 expression and activity in mature neurons is essential for synaptic transmission and complex behaviors including contextual fear memory. Furthermore, Nsun2 robustly shapes affective states associated with depression and anxiety. Bi-directional changes in Nsun2 tRNA methyltransferase activity in adult PFC neurons are associated with directly opposing changes in behavioral despair paradigms and differential effects on anxiety-related behaviors. Consistent with these behavioral studies, our recordings from Nsun2-ablated PFC layer V neurons show significant deficits in mEPSC amplitude and frequencies, which, although a preliminary finding, shows a functional consequence of synaptic protein loss elicited by Nsun2 ablation.

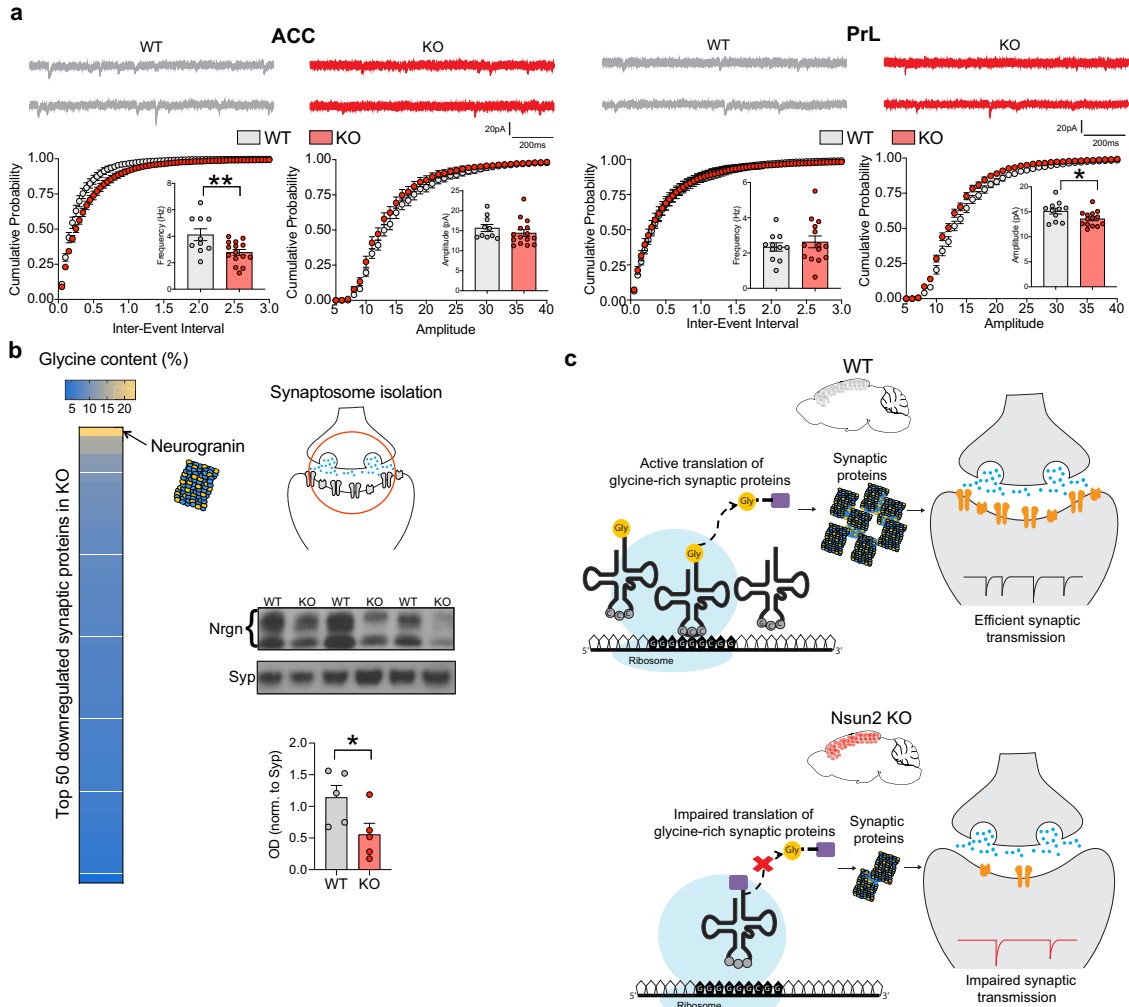

**Fig. 4 Neuronal Nsun2 is essential for synaptic transmission in the prefrontal cortex. a** Electrophysiological recordings from two distinct PFC subregions, ACC (left; $n = 10$ WT neurons, 15 KO neurons) and PrL (right; $n = 11$ WT neurons, 14 KO neurons) in WT and Nsun2 KO mouse brain. mEPSCs were recorded from layer V pyramidal neurons at $-70$ mV (two-tailed $t$ test, **$P = 0.007$, *$P = 0.035$). Data are presented as mean values $+/-$ SEM. **b** Left, glycine content for 50 downregulated synaptic proteins (see Supplementary Table 3) that may contribute to synaptic transmission phenotype, including Neurogranin with 23% glycine. Right, synaptosomes were isolated from Nsun2 KO cortex and immunoblot showed decreased expression of Neurogranin at the synapse in KO cortex ($n = 5$ WT/5 KO; one-tailed $t$ test, *$P = 0.026$). Synaptophysin, which was unaltered in our proteomics screen, was used as a loading control. Data are presented as mean values $+/-$ SEM. **c** Working model for effects of Nsun2 KO on tRNA$^{Gly}$ expression, translation of synaptic proteins, and resulting impairments in synaptic transmission. Source data are available as a Source Data file.

Which molecular mechanisms link Nsun2 to these robust behavioral and physiological phenotypes? Our findings confirm that Nsun2 ablation and overexpression in PFC neurons induce corresponding changes in m5C methylation of Nsun2-regulated cytosine residues positioned in the 3' half of several tRNA iso-acceptor families. These include *Gly* tRNAs harboring altogether four Nsun2-regulated methyl-cytosines including C37 positioned distal from the anticodon hairpin and C46, C47, C48 in the variable loop region, contrasting with a lower number of Nsun2-regulated cytosine residues for other (non-glycinergic) tRNAs in the adult brain. Therefore, it is plausible that the sharp drop in Nsun2-dependent m5C levels in our mutant cortex renders the tRNA$^{Gly}$ family particularly susceptible to degradation or other types of molecular defects resulting in lower levels of expression, which is what we observed for the entire set of 10/10 tRNA$^{Gly}$ isodecoders. Of note, deficits in tRNA$^{Gly}$ levels after germline Nsun2 ablation have been previously reported for the liver[17,35] and skin[10]. The latter finding is interesting in view of Blanco

et al.'s comprehensive mouse tRNA methylome atlas for mouse skin and human dermal fibroblasts[10], which lists only 3/41 iso-acceptor families, or *Gly*, *Glu*, and *Pro*-specific tRNAs, each carrying three Nsun2-methylated cytosine residues (Supplementary Fig. 3b, c). In contrast, brain harbors altogether four Nsun2-methylated cytosine residues for multiple *Gly* isoacceptors including tRNA$^{Gly}_{CCC}$ (ref.[10]) and tRNA$^{Gly}_{GCC}$ (our data), while *Glu* and *Pro* tRNAs in the brain still contain three Nsun2-methylated cytosines (Supplementary Fig. 3c). In addition to highly methylated (>80–90% methylated at baseline) cytosines C46/C47/C48 in the variable loop region, *Gly* tRNA C39 at baseline is moderately methylated (~10–20%), while highly sensitive to changes in Nsun2 methyltransferase activity (Fig. 1c, d). Finally, C37 of tRNA$^{Gly}$ isoacceptors is regulated by a non-Nsun2 RNA methyltransferase, Dnmt2[27]. These findings, taken together, suggest that loss of m5C in the brain is destabilizing to Gly tRNAs that uniquely carry four Nsun2-regulated cytosine residues. Indeed, in the adult mutant cortex with neuron-specific Nsun2

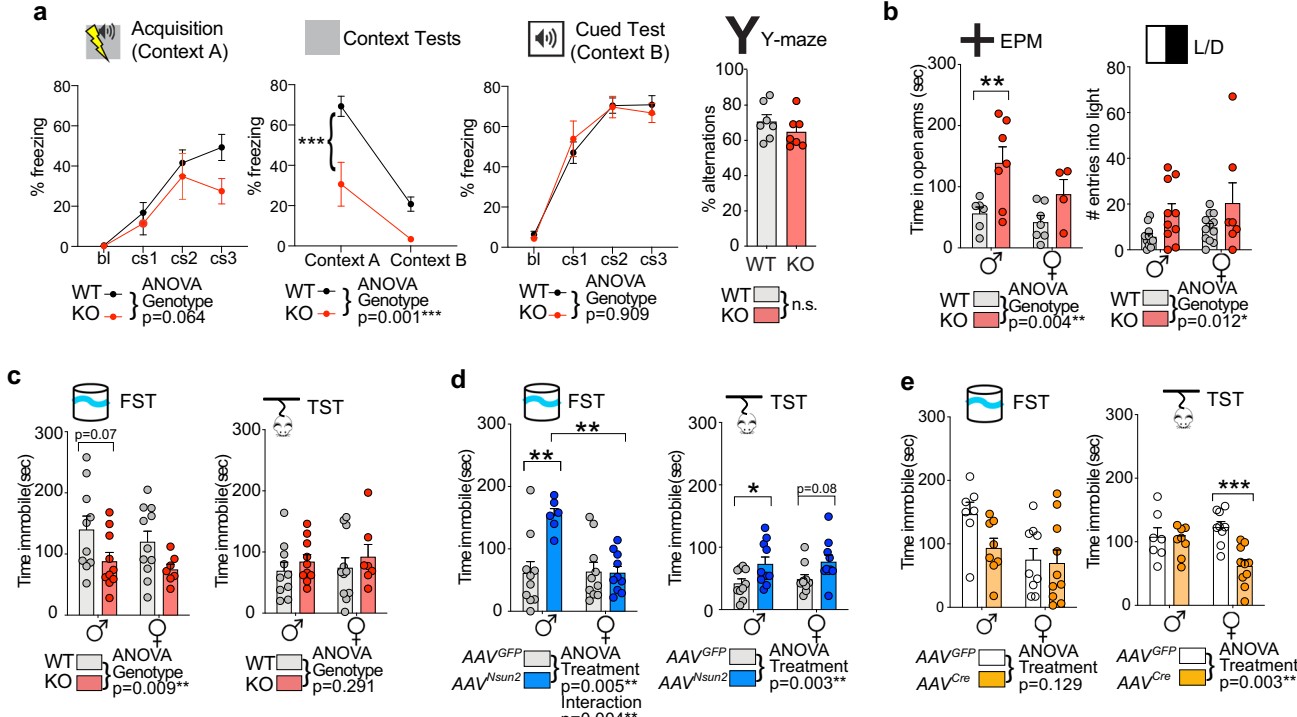

**Fig. 5 Manipulation of neuronal Nsun2 modulates complex behaviors, including cognition and depressive behaviors. a** Behavioral tests of cognition in adult Nsun2 KO and WT mice ($n = 12$ WT, 7 KO). Left, changes in contextual but not cued fear memory in Nsun2 KO mice compared to WT. Right, no changes in working memory were detected in Y-maze spontaneous alternation ($n = 7$/genotype; ***$P < 0.0001$). **b** Nsun2 KO and WT mice in elevated plus maze (EPM; $n = 6$ WT M, 7 WT F, 7 KO M, 4 KO F) and light/dark box (L/D; $n = 11$ WT M, 12 WT F, 10 KO M, 7 KO F; **$P = 0.010$ for KO vs. WT males post hoc comparisons). **c** Nsun2 KO vs. WT mice immobility scores in the forced swim (FST; $n = 10$ WT M, 11 WT F, 10 KO M, 7 KO F) and tail suspension (TST; $n = 10$ WT M, 11 WT F, 10 KO M, 7 KO F). **d** $AAV8^{hSyn1-Nsun2}$GFP vs. $AAV8^{hSyn1}$-GFP control mice immobility scores in the FST ($n = 10$ $AAV^{GFP}$ M, 10 $AAV^{GFP}$ F, 6 $AAV^{Nsun2}$ M, 10 $AAV^{Nsun2}$ F; **$P = 0.002$ for post hoc comparisons) and TST ($n = 10$/sex/condition; *$P = 0.044$ for post hoc comparisons). **e** $AAV8^{hSyn1-Cre}$GFP vs. $AAV8^{hSyn1}$-GFP control mice immobility scores in the FST and TST ($n = 7$ $AAV^{GFP}$ M, 9 $AAV^{GFP}$ F, 8 $AAV^{Cre}$ M, 10 $AAV^{Cre}$ F; ***$P < 0.001$). All data are presented as mean values +/− SEM, and all post hoc $t$ tests are two-tailed with Bonferroni correction. Source data are available as a Source Data file.

deficiency, loss of tRNA expression was specific for the group of *Gly* tRNAs, while no consistent alterations in tRNA isodecoder expression were found for any of the remaining tRNA families. In turn, loss of *Gly* tRNA resulted in a decline in translational efficiencies specifically of Gly-rich (including many synapse-associated) neuronal proteins (Supplementary Table 3 and Supplementary Fig. 4f). Our findings, taken together, point to Nsun2 as a key epitranscriptomic regulator in mature neurons with a critical role in the maintenance of proper levels of *Gly* tRNAs required for the efficient translation of Gly-rich proteins. Disruption of this mechanism then leads to deficits in synaptic transmission and alterations in complex behaviors.

Unexpectedly, our study uncovered a 2.46-fold increase of glycine amino acid levels in the bulk extract from the Nsun2-deficient cortex. In contrast, levels for the remaining 19 canonical amino acids were indistinguishable from the wild-type cortex. Glycine upregulation in the mutant cortex was accompanied by increased protein abundance for enzymes sequentially aligned in glycine biosynthetic pathways starting from a product of glycolysis, 3-phosphoglycerate. Interestingly, however, serine racemase (Srr), a key regulatory enzyme diverting the glycine precursor, L-serine, towards the production of D-serine, was significantly downregulated in the mutant cortex. These findings suggest that the loss of tRNA$^{Gly}$ and the corresponding drop in translational efficiencies of Gly-rich neuronal proteins triggers a compensatory response among multiple key nodes in the glycine biosynthetic pathway resulting in an increase in cortical glycine levels.

Therefore, these data, as an emerging hypothesis, would suggest that mature cortical neurons, when sensing codon-specific disruptions in tRNA supply, may respond by adjusting metabolic and biosynthetic pathways regulating the cellular pool of the cognate amino acids. Alternatively, there is evidence that the metabolism of glycine and its precursor molecule, serine, are part of a general response to translational stress in the brain and in peripheral tissues. For example, mice with a mutation-induced loss of an *Arg*-tRNA gene expressed specifically in the brain, *n-Tr20*, reportedly show ~200% increase of hippocampal glycine and ~100% increase in serine levels in conjunction with upregulated expression of Phgdh (D-3-phosphoglycerate dehydrogenase), an enzyme involved in the early steps of the glycine/serine biosynthetic pathway[43]. Furthermore, a 100–200% increase in serine and glycine metabolism was reported for acute lymphoblastic T-cell leukemia cells carrying a specific ribosomal protein mutation, *RPL10*$^{R98S}$[55]. This effect was thought to result from increased expression of PHGDH, PSAT1, PSPH, and SHMT2[55], comprising the glycine/serine biosynthetic pathway; these proteins also showed upregulated expression in the present study (Fig. 3). Downstream effects of elevated glycine/serine levels could include (i) an increase in the supply of intermediates for purine biosynthesis as a critical building block for nucleic acids, (ii) increased availability of nutrients for the surrounding cells and tissues, and (iii) increased NMDA receptor-mediated excitatory signaling and glycinergic receptor (GlyR)-mediated fast inhibitory

neurotransmission, given that D-serine and glycine are well-established ligands for these two receptor types.

Therefore, in addition to the long-established "glycine neurophysiology" via (1) binding sites on glutamate NMDA receptors[56], and (2) its cognate receptors (GlyR) in subtelencephalic areas[57], our findings point to the third type of mechanism, starting with tRNA$^{Gly}$ epitranscriptomic modification, by which glycinergic pathways could critically regulate adult brain function and complex behaviors. Pharmacological alterations in glycine function in the brain[58] have shown promise for alleviating cognitive deficits[59] and depressive symptoms in humans[60] and animal models[61,62]. The findings presented here broadly align with the therapeutic promise of glycinergic pathways. Thus, our Nsun2 mutant model for tRNA m$^5$C epitranscriptomic modification unveils a selective vulnerability of glycine-specific tRNA isodecoders, ultimately resulting in decreased translational efficiencies of glycine-rich proteins in the affected neurons, in conjunction with altered glutamatergic synaptic signaling in two distinct subregions of the PFC (ACC and PrL; Figs. 2c and 4a, b) and antidepressant-like behavioral phenotypes (Fig. 5c, e). Importantly, the novel fast-acting antidepressant drug ketamine exerts therapeutic effects via rapid modulation of protein synthesis, including translational initiation[3] and elongation[2], and mTOR signaling[57], providing an additional link between translational regulation of the proteome and the neurobiology and treatment of depression and anxiety. Finally, we note that while tRNAs are the main and most well-characterized target of Nsun2 methyltransferase activity, additional work will be required to explore the heterogenous non-tRNA species potentially methylated by Nsun2, including small ribonucleoprotein-(vault)-associated RNAs[24,63], microRNAs[64], and even a limited subset of messenger RNAs[65–67]. Future work will further investigate additional Nsun2 targets in the adult brain as well as tRNA-induced proteomic changes that may produce differential effects on glutamatergic signaling in PFC subregions. Our work here adds to the existing literature on epitranscriptomic mechanisms in the mature brain, which has until now solely focused on adenosine methylation (m6A) on mRNA, a modification which regulates mRNA stability, splicing, and localization, and provided causal links between m$^6$A, brain function, and behavioral outcomes, including learning, memory, and neuronal function[68–71]. We now show evidence that the m$^5$C modification on tRNAs, specifically tRNA$^{Gly}$, is another level of epitranscriptomic regulation for neurobiology and behavior in the mature brain.

## Methods

**Animals**. All animal work was approved by the Institutional Animal Care and Use Committee of the Icahn School of Medicine at Mount Sinai. Mice were group-housed 2–5/cage with ad libitum access to food and water and a 12 h light/dark cycle (lights off at 7 pm) under constant conditions (21 ± 1 °C; 60% humidity). Mice bred in-house were weaned at ~P28, housed with same-sex littermates, and ear-tagged/genotyped.

**Nsun2 conditional knockout mice**. C57BL/6N-Nsun2$^{tm1c(EUCOMM)WtsiOulu}$ mutant mouse sperm were obtained from the Wellcome Trust Sanger Institute (Cambridgeshire, UK). This line is modified from EUCOMM ES clone EPD0105_2_F10, after breeding with a Flp recombinase deleter line to convert the original knockout first targeted allele (tm1a) into a conditional allele (tm1c). Mice with the tm1c allele exhibit a phenotypical wild-type state although exon 6 of Nsun2 is flanked by loxP sites, where the presence of Cre-recombinase excises this exon and produces a frame-shift, resulting in early termination of Nsun2 translation (Fig. 1a). IVF was performed with C57BL/6N-Nsun2$^{tm1c(EUCOMM)WtsiOulu}$ mutant mouse sperm and wild-type (WT) C57BL/6N females. Mice were then bred further to obtain Nsun2$^{2lox/2lox}$ mice (two copies of tm1c mutant allele). Nsun2$^{2lox/2lox}$ mice were crossed with a CamK-Cre$^+$ line to produce CamK-Cre$^+$,Nsun2$^{2lox/2lox}$ mice for knockout of Nsun2 in excitatory forebrain neurons. This particular calmodulin-kinase II (CamK)-Cre transgenic line is associated with widespread neuronal Cre-mediated deletion across the forebrain before postnatal day 18, as previously described[73–76]. For behavioral and molecular experiments, we bred

CamK-Cre$^-$,Nsun2$^{2lox/2lox}$ males with CamK-Cre$^+$,Nsun2$^{2lox/-}$ females and obtained expected Mendelian ratios of offspring genotypes ($X^2$(3, $N = 99$) = 0.459, $P = 0.928$). Age- and sex-matched CamK-Cre$^-$,Nsun2$^{2lox/2lox}$ or Nsun2$^{2lox/-}$ were used as wild-type controls in genotype experiments. Mice were 13–16 weeks old for all behavioral experiments and sacrificed at 13–16 weeks for all molecular experiments except for tRNA bisulfite sequencing of CamK-Cre$^+$,Nsun2$^{2lox/2lox}$ mice, which were ~8 weeks old at the time of sacrifice.

**Viral microinjections**. For viral microinjection surgeries into adult mouse PFC (AAV8$^{hSyn1-Nsun2GFP}$, AAV8$^{hSyn1-GFP}$, AAV8$^{hSyn1-CreGFP}$), 10–12 week-old C57BL/6J mice were anesthetized with isoflurane and 1 µl of virus per hemisphere (bilateral injection) was injected at a rate of 0.25 µl/min using a Hamilton syringe (Reno, NV), a micropump (Stoelting) and a stereotactic frame (Stoelting). Stereotactic coordinates for injection were as follows: 1.5 mm anterior/ posterior, ±0.5 mm medial/lateral, and 1.5 mm dorsal/ventral. Control animals received 1 µl per hemisphere of AAV8$^{hSyn1-GFP}$ using the same conditions.

**Immunohistochemistry**. Adult mice were anesthetized with a terminal intraperitoneal injection of a ketamine/xylazine mixture (IP: 200 and 30 mg/kg, respectively). Transcardial perfusion was performed with 100 ml of 10% sucrose followed by 200 ml of 4% paraformaldehyde in PBS. Brains were removed and placed in 4% formaldehyde overnight at 4 °C, followed by incubation in 30% sucrose until isotonic. After embedding in OCT compound (Tissue-Tek), the brains were cut on a freezing microtome (Leica SM2010 R) into 30-µm coronal sections and placed in 1× PBS. Antibody staining was performed as follows: coronal sections containing prefrontal cortex were blocked and permeabilized with 10% BSA and 0.05% Triton X-100 in 1× PBS for 1 h at room temperature, followed by incubation in primary antibody (NeuN, 1:200) diluted in 0.01% Triton X-100 overnight at room temperature. The sections were washed for 5 min in PBS followed by incubation in secondary antibodies diluted 1:5000 in PBS for 1 h at room temperature. Sections were washed briefly in PBS before being mounted on Superfrost Plus slides (Fisher) with DAPI Fluoromount-G media (SouthernBiotech). Imaging was done using a Zeiss CLSM780 upright microscope. Image processing was done in NIH ImageJ software.

**BaseScope in situ hybridization**. Adult CamK-Cre$^+$,Nsun2$^{2lox/2lox}$ and WT mice (~2–3 months old) were anesthetized with a terminal intraperitoneal injection of a ketamine/xylazine mixture (IP: 200 and 30 mg/kg, respectively). Transcardial perfusion was performed with 100 ml of 10% sucrose followed by 200 ml of 4% paraformaldehyde in PBS. Brains were removed and placed in 4% formaldehyde overnight at 4 °C, followed by incubation in 30% sucrose until isotonic. After embedding in OCT compound (Tissue-Tek), the brains were cut on a freezing microtome (Leica SM2010 R) into 10-µm coronal sections and placed in 1× PBS until ready to use.

BaseScope (ACDBio) in situ hybridization for Nsun2 mRNA was performed using a single ZZ probe targeting nucleotides 615–659 of Nsun2 mRNA, which exclusively targets half of exon 6 (the exon deleted in our Nsun2 KO mouse). Briefly, sections were slide-mounted and dried for 1 h at 60 °C. A hydrophobic barrier was drawn around the tissue sections before incubation in hydrogen peroxide and protease. The ZZ probe was added to the sections for 2 h at 40 °C. The slides were incubated in a series of amplifier oligonucleotide sequences, with wash steps in between. The signal was developed with BaseScope FAST Red for 10 min at room temperature, followed by hematoxylin staining. The sections were coverslipped with VectaMount mounting media (Vector Labs). Images were acquired on a Zeiss Axio Imager Z2 Upright microscope configured with a ×63 objective.

**Subcellular fractionation for synaptosomes**. Frozen cortical tissue was homogenized in ice-cold HEPES-sucrose with a Teflon homogenizer at 600 rpm and centrifuged at 1000×g at 4 °C for 10 min. The supernatant was removed and centrifuged two more times until no pellet was visible. The supernatant was then centrifuged at 10,000×g for 10 min, and the supernatant was removed and kept as the cytoplasmic fraction. Pelleted material was used as the crude synaptosomal fraction and resuspended in HEPES-sucrose, followed by two additional washes with HEPES-sucrose. After the final wash, the synaptosome pellet was resuspended in 100 µl RIPA buffer and quantified by BCA assay for western blotting.

**Western blotting**. Cortical tissue was homogenized using a vibrating pestle in buffer containing SDS, HEPES, sucrose, and protease inhibitors and quantified using a BCA assay. In total, 30 µg of purified protein (10 µg for synaptosomes) was denatured at 95 °C for 10 min and then electrophoresed on 4–12% NuPAGE Bis-Tris protein gels (Invitrogen) in Novex SDS running buffer (Invitrogen). Gels were then transferred to nitrocellulose membranes using the Trans-blot Turbo Transfer System (Bio-Rad), and efficient transfer was confirmed with direct-blue staining. Membranes were then incubated with blocking buffer (5% milk) followed by overnight incubation at 4 °C with the primary antibody in blocking buffer. Primary antibodies were Nsun2 (Proteintech, 1:1000), β-actin (Cell Signaling Technology, 1:5000), H3 (Novus Biologicals, 1:10,000), Neurogranin (Santa Cruz Biotechnology, 1:1000), and Synaptophysin (Abcam, 1:500). Membranes were washed and

incubated with a peroxidase-labeled secondary antibody (rabbit: 1:10,000; mouse: 1:5000) for 1 h, followed by another set of washes. Bands were visualized with Immobilon Western Chemiluminescent HRP Substrate (Millipore), exposed, and developed before quantification with NIH ImageJ software. Band size was normalized to actin or H3 (or synaptophysin for synaptosome blot) to control for equal loading and protein concentration. An unpaired one- or two-tailed $t$ test was used to compare KO and WT and significance was denoted at $P < 0.05$.

**RNA sequencing**. For RNAseq, the total RNA was extracted from Nsun2 KO or WT cortex using the Directzol RNA MiniPrep Kit (Zymo Research) including on-column DNase treatment, and quantity was measured on Qubit fluorometer. The quality of the total RNA was checked on the Agilent Bioanalyzer using the RNA 6000 Pico Kit. In total, 5 μg of total RNA was rRNA depleted using the Illumina RiboZero Gold rRNA depletion kit according to the manufacturer's instructions. We then used 40–50 ng rRNA-depleted RNA samples as input for the Smartr Stranded RNAseq kit (Takara Biosystems) following the manufacturer's instructions for library preparation and amplifying libraries with 11 PCR cycles. Four libraries (2 KO, 2 WT) were then pooled at equimolar concentrations in one lane for sequencing on the Illumina HiSeq at the New York Genome Center.

*Data analysis.* After paired-end sequencing, samples were mapped to mouse (GRCm38.p5_M13) with STAR (v2.5.3a) using a two-method step protocol following tool specifications[77]. Samples were counted by exon using featureCounts (subread v.1.5.2).

**tRNA bisulfite sequencing**. The tRNA amplicon bisulfite sequencing protocol was adapted from Bormann et al.[78] (Supplementary Fig. 2a). Cortex or PFC punches were homogenized in Trizol reagent (Invitrogen) and RNA was extracted, purified, DNase-treated, and quantified using the Qubit fluorometer. We used the Agilent Bioanalyzer Small RNA kit to confirm the presence of tRNAs after total RNA extraction. In total, 1 μg of the total RNA was used for bisulfite conversion with the EZ RNA Methylation kit according to the manufacturer's instructions (Zymo Research). Due to the strong secondary structure of tRNAs, the PCR step during bisulfite conversion was repeated three times consecutively, as previously reported[10] to completely denature tRNAs. Following bisulfite treatment and cleanup, we performed cDNA synthesis using the reverse primer for each of our six tRNA isodecoders (i.e., *GlyGCC-1-1*[35], *GluTTC-1-1, GluCTC-1-1, AspGTC-1-1*[35,78], *ValAAC-2-1*[35], *ProTGG-2-1;* primer sequences listed in Supplementary Table 7). Targeted cDNA was then PCR-amplified with the tRNA-specific forward primer and products were size-verified on a 4% agarose gel. PCR amplicons (~120 nt including tRNA amplicon and overhangs for adapter ligation) were then extracted and purified (Qiaquick Gel Extraction Kit, Qiagen). We labeled each amplicon with a unique index (Nextera XT, Illumina) and pooled up to 20 amplicons in equimolar concentrations before running with 75 bp paired-end reads on the MiSeq (Illumina). Read depth for each amplicon is presented in Supplementary Fig. 2c.

*Data analysis.* Data were analyzed using bisAmp, a publicly available web-based pipeline to identify and quantify methylated cytosine at tRNA targets[78]. tRNA bisulfite sequencing data from bisAMP (methylation percentages for each cytosine) was further analyzed by conducting a $t$ test for each cytosine with FDR correction using the two-stage linear step-up procedure of Benjamini, Krieger, and Yekutieli.

*Bisulfite conversion positive control.* To confirm the efficiency of RNA bisulfite conversion, pure human RNA (from a previous study) was bisulfite-converted as described above and traditional cDNA synthesis (Quantitect Reverse Transcription kit, Qiagen) was performed followed by SYBR-based PCR amplification and Sanger sequencing of the 28S rRNA (Supplementary Fig. 2b).

**tRNA sequencing**. RNA was extracted as previously described, and 5 μg of total DNase-treated RNA was used for pre-treatment and library prep. Pre-treatment of RNA, library preparation, next-generation sequencing, and some aspects of data analysis were performed by Arraystar, Inc. (Rockville, MD). Full-length tRNA sequencing and tRNA fragment sequencing were performed separately to provide more accurate size selection and better sequencing depth and coverage.

*Full-length tRNA sequencing.* For full-length tRNA sequencing, 5 μg of total DNase-treated RNA was resolved on a urea-polyacrylamide gel and recovered within a size window of 60–100 nt and then pre-treated with demethylation enzymes[32] to prepare for efficient reverse transcription and adapter ligation. Specifically, we coupled nucleotide demethylation and the well-established Hydro-tRNAseq[31] to optimize efficiency in tRNA seq library preparation, which included (1) m1A and m3C demethylation of purified tRNA, (2) limited alkaline hydrolysis of demethylated tRNA by carbonate buffer, and (3) calf intestinal phosphatase (CIP) dephosphorylation and then T4 polynucleotide kinase (PNK) re-phosphorylation of the partially hydrolyzed tRNA fragments. Following this pre-treatment, resulting fragments of ~19–35 nt were converted to small RNA sequencing libraries using NEBNext Multiplex Small RNA Library Prep Set (Illumina) and quantified on the Agilent Bioanalyzer before 50 bp single-end sequencing on the Illumina NextSeq 500 instrument.

*Data analysis.* Sequencing quality was assessed with FastQC, and raw sequencing read data that passed the Illumina chastity filter was used for further data analysis, including alignment of trimmed reads (cutadapt) to mature tRNA reference sequences from GtRNAdb[79] and mitotRNAdb[80] using BWA. For tRNA alignment, the maximum number of mismatches were two. tRNA expression levels were measured with tag count and the multi-map-corrected number of reads overlapping the tRNA (mrcount) was then calculated. For differential expression analysis, we used the R package edgeR, with a fold change cutoff of 1.5 and requirement that CPM >= 20 (mean in one group) and generated $P$ values and $q$ values (FDR-adjusted $P$ values) based on results of the exact test by the negative binomial distribution. Significantly up- or downregulated tRNAs were denoted if $q < 0.05$.

*tRNA fragment/half (tiRNA/tRF) sequencing.* For tRNA fragment/half sequencing, 5 μg of total DNase-treated RNA was pre-treated for m1A and m3C demethylation, 3' deacylation and 3' – cP removal for 3' adapter ligation, and 5' phosphorylation for 5' adapter ligation. Pre-treated total RNA was then ligated to 3' and 5' small RNA adapters and cDNA was synthesized and amplified using Illumina's Small RNA RT primers and amplification primers. Following amplification, ~134–160 bp PCR products were extracted and purified from a PAGE gel and resulting libraries were quantified on the Agilent 2100 Bioanalyzer. tRNA fragment libraries were denatured and diluted to a loading volume of 1.3 ml and loading concentration of 1.8 pM for loading onto a reagent cartridge and sequencing on the Illumina NextSeq 500 system using NextSeq 500/550 V2 Kit (#FC-404-2005, Illumina) according to the manufacturer's instructions.

*Data analysis.* Raw sequencing read data that passed the Illumina chastity filter was used for further data analysis. Reads were trimmed and aligned to mature tRNA reference sequences from GtRNAdb and mitotRNAdb allowing for 1 mismatch, and reads that did not map were then mapped to precursor tRNA sequences (pre-tRNAs) allowing for 1 mismatch with bowtie software. Expression of mapped tRNA fragments was measured with CPM, which were summed across isodecoders for each fragment type and individual $t$ tests were performed for each isoacceptor. Any fragment that had a CPM of 0 for any sample was excluded from the analysis. Statistical significance was denoted by $P$ value < 0.05.

**YAMAT/UMI seq**. Y-shaped Adapter MAture tRNA (YAMAT) sequencing was adapted from Shigematsu et al.[33] to include unique molecular identifiers (UMIs) in order to avoid overamplification of individual tRNA isodecoders due to modification levels or abundance. In total, 5 μg total RNA was deacetylated to remove amino acids from 3' ends and demethylated to remove m1A, m1C, and m3C modifications using a proprietary demethylation mix (Arraystar, Inc). In total, 40 μm of YAMAT forked linkers (Y-3-AD_UMI: 5'-P-GTATCCAGTNNNNTGG AATTCTCGGGTGCCAAGG-3'-ddC; Y-5-AD_UMI: 5'-GTTCAGAGTTCTAC AGTCCGACGATCNNNNACTGGATACTGrGrN-3') were then incubated with pure demethylated and deacetylated RNA followed by the addition of 10× annealing buffer (50 mM Tris HCl pH 8, 100 mM MgCl$_2$, 5 mM EDTA) and then overnight incubation with T4 RNA ligase 2. Linker-ligated RNA was then incubated with RT Primer (TruSeq Small RNA Library Prep Kit, Illumina), and reverse transcription was performed with Superscript III RT (Invitrogen) followed by bead purification. Libraries were amplified using Phusion Hotstart II Polymerase (Thermo Scientific) with primers and indexes from the TruSeq Small RNA kit (Illumina) for 11 PCR cycles. Amplified libraries were then bead-purified and run on the Agilent Bioanalyzer for confirmation of library size and quantified with the Qubit Fluorometer (Invitrogen). Seven libraries ($n = 3$ WT, 4 KO) were pooled at equimolar concentrations and run with 75 bp paired-end reads on the MiSeq (Illumina).

*Data analysis.* Raw sequencing reads were processed using cutadapt to trim adaptor sequences and UMItools to extract UMIs from reads and perform deduplication. Paired-end reads were merged using Pear and differential expression analysis was performed using Deseq2, with significant differences in KO vs. WT tRNA expression denoted at adjusted $P < 0.05$.

**Real-time qPCR**. For real-time qPCR to identify *Nsun2* mRNA expression levels after knockout or overexpression, total RNA was extracted from cortex (KO) or PFC punches (viral overexpression samples) that were identified with a *Bluestar* GFP flashlight[81] (Nightsea) from 1-mm coronal brain sections. RNA was prepared with the Directzol RNA MiniPrep Kit (Zymo Research) with on-column DNase treatment and reverse transcribed with the Quantitect Reverse Transcription kit (Qiagen) before being subjected to Taqman qPCR using Taqman Universal Master Mix and Taqman probes for *Nsun2* (Assay ID: Mm01349532_m1) and *Gapdh* as a housekeeping gene (Assay ID: Mm99999915_g1; Applied Biosystems). Data were analyzed using the comparative Ct method and normalized to the housekeeping gene and AAV$^{hsyn1-GFP}$ or WT same-sex littermate controls. Unpaired $t$ tests were used to compare experimental groups and significance was denoted at $P < 0.05$.

*tRNA/tRF expression.* To further verify full-length tRNA expression after next-generation sequencing, we reverse-transcribed 1 μg of total RNA and amplified

target tRNAs using pre-validated PCR primer sets specific to mature tRNA sequences (*GlyGCC, AspGTC*; Arraystar, Inc,) using SYBR-based qPCR. For full-length *GluTTC* and 5' tRNA fragments/halves (*GlyGCC, GluTTC, GluCTC*), we reverse-transcribed total RNA using a sequence-specific small RNA RT-stem-loop primer and used qPCR to amplify target tRNA/tRFs using custom-designed Taq-man probes (Applied Biosystems). Full-length tRNA and tRF amplicons were run on a 4% agarose gel to confirm the size. All qPCR data were analyzed using the comparative Ct method and normalized to the housekeeping gene 5S rRNA and WT same-sex, littermate controls. A one-sample *t* test was used to compare KO to WT.

**tRNA charging and non-$m^5C$ tRNA modification assays**. tRNA charging analysis and measurement of non-$m^5C$ modifications were performed using DM-tRNAseq as previously described[32,82,83] with minor modifications as follows. For tRNA charging studies, tRNAs were subjected to a one-pot periodate oxidation/β-elimination procedure (adapted from ref.[83]), followed by a ligation reaction and RNA-seq library construction. Briefly, up to 500 ng of total RNA from Nsun2 KO and WT cortex was eluted in 7 µl and mixed with 1 µl of 90 mM sodium acetate buffer, pH 4.8. Next, 1 µl of freshly prepared 150 mM sodium periodate solution was added for a reaction condition of 16 mM NaIO4, 10 mM NaOAc pH 4.8. Periodate oxidation proceeded for 30 min at room temperature. Oxidation was quenched with the addition of 1 µl of 0.6 M ribose to 60 mM final concentration and incubated for 5 minutes followed by the addition of 5 µl freshly prepared 100 mM sodium tetraborate, pH 9.5 for a final concentration of 33 mM. This mixture was incubated for 30 min at 45 °C. To stop β-elimination and perform 3' end repair, 5 µl of T4 PNK mix (200 mM Tris HCl pH 6.8, 40 mM MgCl2, 4 U/µl T4 PNK, from New England Biolabs) was added to the reaction and incubated at 37 °C for 20 min. T4 PNK was heat-inactivated by incubating at 65 °C for 10 min. This 20 µl reaction mixture was used directly in the adaptor ligation reaction. tRNA libraries were then prepared[32] (with or without AlkB demethylase treatment to increase efficiency and quantitation and to investigate modification) and Illumina sequencing was performed. Data were aligned using Bowtie to a modified mouse tRNA genome file containing chromosomal-encoded tRNAs and mitochondrial-encoded tRNAs to determine charging levels and to identify potential modification misincorporations and modification fractions.

**Ribosome profiling**. RiboSeq libraries were constructed as follows, which was adapted from Ingolia and colleagues[44]. Cortical tissue was homogenized in lysis buffer (containing cycloheximide) and cleared for 10 min at 20,000×*g* at 4 °C. Monosomes obtained by RNase I digestion were purified using spin-column chromatography. Briefly, MicroSpin S-400 HR columns (Cytiva) were equilibrated with 3 ml of Polysome buffer by gravity flow, then spun down 4 min at 600×*g*. In total, 100 µl of RNase I digested lysate was loaded onto the column and spun for 2 min at 600×*g*. The flow-through was collected, and the RNA was extracted using RNA Clean & Concentrator-25 kit (Zymo). The rRNA depletion was performed onto linker ligated RPFs using riboPOOLs (siTOOLS). RNAseq libraries were constructed using SMARTer-Stranded Total RNA-Seq Kit v2 - Pico Input Mammalian (Takara), as recommended by the manufacturer. RiboSeq and RNAseq libraries were sequenced on HiSeq4000 (Illumina) at the Center for Advanced Technology (UCSF).

*Data analysis*. The adapters in the sequencing reads were removed using cutadapt (v3.1) with options "—trimmed-only -m 15 -a AGATCGGAAGAGCAC". The PCR duplicates in the reads were collapsed using CLIPflexR v0.1.19. The Unique Molecule Identifiers (UMIs) for each read were extracted using umi_tools v1.1.1 with the options "extract—bc-pattern=NN" for the 5' end and options "extract—3prime—bc-pattern=NNNNN" for the 3' end. Reads corresponding to rRNA and other non-nuclear mRNA were removed by aligning out the reads using Bowtie2 v2.4.2 on a depletion reference (rRNA, tRNA, and mitochondrial RNA sequences). This depletion reference was built from the noncoding transcriptome for Mus musculus (Ensembl release v96). The reads that did not align to the depletion reference were aligned to the Mus Musculus mRNA transcriptome using Bowtie2 with options "—sensitive—end-to-end—norc". The mRNA transcriptome was built using the cDNA longest CDS reads of *Mus musculus* downloaded from Ensembl release v96. The resulting reads were converted to bam files and then sorted using samtools v1.11. The duplicate reads in the sorted files were removed using umi_tools v1.1.1 with options "dedup".

The quality check and downstream processing of the processed reads were performed using Ribolog v0.0.0.9. After quality check, the P-site information was converted to codon counts in Ribolog. The codon reads resulting from stalling were corrected using the CELP (Consistent Excess of Loess Preds) method in Ribolog. After median normalization and removal of transcripts with 0 count, the translational efficiency testing was performed in Ribolog as well.

**Mass spectrometry (LC-MS/MS)**

*Digestion*. PFC tissue was homogenized using a vibrating pestle in buffer containing SDS, HEPES, sucrose, and protease inhibitors and quantified using a BCA assay. Proteins were purified by Wessel/Flügge extraction[84], and the resulting protein pellets were solubilized in 8 M urea, 50 mM triethylammonium bicarbonate

(TEAB), 10 mM dithiothreitol (EMD Chemicals), and disulfide bonds were reduced at room temperature for 1 h. Thiols were alkylated using 20 mM iodoacetamide (Sigma) for 1 h at room temperature in the dark. Urea was diluted using 50 mM TEAB and proteins were digested, first with lysyl endopeptidase (Wako Chemicals) for 3 h and then overnight with sequencing grade modified trypsin (Promega). Digestion was halted by the addition of TFA. Peptides were purified by solid-phase extraction using Oasis HLB cartridges (Waters) according to the manufacturer's specifications.

*Isotopic labeling*. Peptide pellets were redissolved in 100 mM TEAB and labeled with tandem mass tag 11-plex (TMT, Thermo Scientific) dissolved in anhydrous acetonitrile. Labeling proceeded for 1 h at room temperature and was quenched by the addition of hydroxylamine (Thermo Scientific). Based on a label-check, the labeled peptides were mixed in equal amounts.

*High-pH reverse-phase fractionation*. Peptides were pre-fractionated using a Dionex 3000 Ultimate loading pump equipped with a 2.1 × 150 mm 3.5 µm Xbridge C18 column (Waters). Solvent A consisted of 10 mM ammonium hydroxide (Sigma-Aldrich) in water, pH 10 and solvent B consisted of 10 mM ammonium hydroxide, 90% acetonitrile (ACN) in water, pH 10. Peptides were fractionated across a 60-min gradient, and collected fractions were concatenated to yield a total of 21 fractions.

*LC-MS/MS*. Fractionated peptides were separated using a Dionex 3000 Ultimate HPLC equipped with a NCS3500RS nano- and microflow pump (Dionex). Trapping and separation were carried out using a 100 µm × 20 mm Acclaim PepMap C18 trap column (Thermo Scientific) and a 75 µm × 120 mm pulled-emitter nanocolumn (Nikkyo Technos), respectively. Solvent A was 0.1% formic acid in water and solvent B was 0.1% formic acid, 80% ACN in water. All LC-MS solvents were of Optima grade and purchased from Fischer. The HPLC was connected to a Q-Exactive HF mass spectrometer (Thermo Scientific) operating in positive ion DDA mode. The MS2 resolution was set to 60 k.

*Data analysis*. Raw data were searched using MaxQuant v. 1.6.6.0[85] using standard settings. Spectra were queried against the *Mus musculus* proteome (obtained from uniprot.org on February 2019, 54185 sequences) with a false discovery rate (FDR) of 1% applied on PSM, peptide, and protein level. Ratio compression from co-fragmentation was minimized by requiring a minimum peptide interference factor (PIF) of 0.75. Further data analysis was performed within the Perseus framework[86]. *T* test difference (KO/WT) was calculated by averaging signals for all KO samples and subtracting (because the data is log2 transformed) the average signal from all WT samples. An FDR-corrected Student's *t* test (FDR = 1%) was used to test for significant changes between the two conditions. The standard deviation was calculated by selecting three random, but non-repeating replicates from each group and taking the average from them. A simple standard deviation was calculated between the two averages.

**Metabolomic profiling of amino acids**. For unbiased metabolomic profiling of amino acids, 10–20 mg of fresh frozen cortical tissue was homogenized on ice in 80% methanol with heavy amino acid mix standards (Cambridge Isotope Laboratories MSK-A2-1.2 mix). After centrifugation at 4 °C, the supernatant was placed into a new tube and dried using a SpeedVac before storage at −80 °C for later processing.

*Pre-sample analysis*. Dried polar samples were resuspended in 60 µl pre-chilled 50% acetonitrile, vortexed for 10 s, centrifuged for 10 min at 4 °C, and 13,200 resolution per minute, then 5 µl from each sample was transferred to create a pooled sample. This pooled sample was further diluted with 1:3 and 1:10 dilution factors, 5 µl of 1:1, 1:3, 1:10 dilution were injected onto LC-MS system from low to high concentration to get a dynamic range for detection. Polar metabolites were separated on a SeQuant® ZIC®-pHILIC 5 µm polymer (150 × 2.1 mm) column (EMD Millipore) connected to a Thermo Vanquish ultra high pressure liquid chromatography instrument coupled to a Q Exactive Plus Hybrid Quadrupole-Orbitrap mass spectrometer (Thermo Fisher Scientific) with a heated electrospray ionization source. Chromatographic separation was achieved by mixing mobile phase A consisted of 20 mM ammonium carbonate with 0.1% (vol/vol) ammonium hydroxide (adjusted to pH 9.3 with formic acid) and mobile phase B of acetonitrile in the following gradients: 90–40% B (0–22 min), held at 40% B (22–24 min), 40–90% B (24–24.1 min), and reequilibrated at 90% B (24.1–30 min) at a flow rate of 0.15 ml/min. Mass spectrometric data were acquired in polarity switching mode for both MS1 (full MS) and MS2 (data-dependent acquisition) with the following parameters: spray voltage, 3.0 kV; capillary temperature, 275 °C; source temperature, 250 °C; sheath gas flow, 40 a.u.; auxiliary gas flow, 15 a.u. The full MS scans were acquired with 70,000 resolution, 1 × 10⁶ ACG target, 80 ms max injection time, and a scan range of 55–825 *m/z*. The data-dependent MS/MS scans were acquired at a resolution of 17,500, 1 × 10⁵ ACG target, 50 ms max injection time, 1.6 Da isolation width, stepwise normalized collision energy of 20, 30, and 40 units, with 8 s dynamic exclusion and a loop count of 2. Relative quantification of amino acids (heavy and endogenous) was performed in Skyline Daily (v.20.2.1.315) with

the maximum mass error and retention time tolerance set to 2 ppm and 12 s, respectively, referencing in-house retention time for amino acids.

*Sample analysis.* On the same day, after acquiring data for detection of amino acids, a 30 μl sample volume was transferred to another centrifuge tube accordingly, and samples were diluted further with 1:3.8 dilution factor, given the final resuspension volume of 114 μl. From each sample, 14 μl was transferred to the sample vial and 14 μl was transferred to create another pooled sample. This pooled sample was also diluted with 1:3 and 1:10 dilution factors. Both samples and pooled samples were injected onto LC-MS system, as described above, with the same settings at 5 μl. Relative quantification of amino acids (heavy and endogenous) was again performed in Skyline Daily (v.20.2.1.315) in the same manner. The pooled samples data from both pre-sample analysis and sample analysis were employed for the heavy amino acids calibration curve describe below.

*Data analysis.* Amino acid levels in the samples (i.e., endogenous) were normalized to the averaged factor of heavy amino acids against the pooled sample at 1:1 dilution from sample analysis data. The endogenous amino acids were further normalized to sample tissue weights by setting one of the sample weights as a factor of 1. Conversion of relative quantification of amino acid levels to concentration was done using simple linear regression formula such as, $y = \alpha + \beta x$; where $y$ denotes integrated MS1 area, $\alpha$ denotes $y$ intercept and $\beta$ denotes slope, hence solving $x$ for concentration. Cysteine was the only amino acid (of the 20 canonical amino acids) that was unable to be measured due to high reactivity. Amino acid level fold changes between KO and WT were calculated and $P$ values were obtained using individual unpaired two-tailed $t$ tests for each amino acid with FDR correction using the two-stage linear step-up procedure of Benjamini, Krieger, and Yekutieli without assuming a consistent SD and with $q = 1\%$.

*Heavy amino acids calibration curve.* Heavy amino acid metabolite levels of pooled samples were arranged from low to high concentration from pre-sample analysis and sample analysis data yielding eight concentrations; 1.10, 3.29, 4.17, 10.96, 12.50pmol, 21.93, and 41.66 pmol. Simple linear regression was employed to determine $\alpha$ and $\beta$ of each heavy amino acid, only those concentrations (at least five points) that gave the "best" R-squared value were used to generate a prediction of endogenous amino acid concentrations.

**Codon content analysis.** To estimate changes in translation efficiency in response to Nsun2 knockdown, for every gene, we normalized changes in protein expression by changes in RNA levels. We used the resulting value (logTER, log of translation efficiency ratio), to assess the extent to which codon and/or amino acid contents were associated with translation efficiency. For this, we performed two analyses: (i) for codons or amino acids of interest, we calculated the Pearson correlation coefficient and its associated $P$ value, and (ii) we selected the genes in the top quartile of codon or amino acid contents and perform gene-set enrichment analysis using our iPAGE package[87]. For a given gene-set, iPAGE reports a mutual information value and its associated $z$ score along with a heatmap visualizing the enrichment/depletion pattern across the range of the input values (in this case, logTER).

**Slice electrophysiology.** Nsun2 KO and WT mice (~3 months old) were sacrificed and their brains rapidly removed and placed in saturated (95% $O_2$ and 5% $CO_2$), ice-cold artificial cerebrospinal fluid (ACSF) containing 126 mM NaCl, 2.5 mM KCl, 2.5 mM $CaCl_2$, 1.2 mM $MgCl_2$, 25 mM $NaHCO_3$, 1.2 mM $NaH_2PO_4$, and 11 mM D-glucose. Coronal slices (300 μm) containing the medial prefrontal cortex (mPFC) (1.54–2.8 mm anterior to the bregma) were sectioned on a Leica VT1200 Vibratome and immediately transferred into an incubation chamber (Harvard Apparatus) containing oxygenated ACSF. Slices were recovered in the chamber for at least 1 h at room temperature (21–23 °C) before recordings. Slices were then transferred to a recording chamber and continuously perfused with heated (32 °C) ACSF equilibrated with 95% $O_2$ and 5% $CO_2$. All recordings were performed with 100 μM picrotoxin in the ACSF to block GABA$_A$ receptor-mediated inhibitory synaptic responses. Glass recording pipettes (3–4 MΩ) were filled with a solution containing 142 mM Cs-gluconate, 8 mM NaCl, 10 mM HEPES, 0.4 mM EGTA, 2.5 mM QX-314 [N-(2,6-dimethylphenylcarbamoylmethyl) triethylammonium bromide], 2 mM Mg-ATP, and 0.25 mM GTP-Tris, pH 7.25. Data were collected with a MultiClamp 700B amplifier (Molecular Devices), digitized with a Digidata 1322 A data acquisition system (Molecular Devices), and analyzed with pClamp Software (version 10.7; Molecular Devices).

*Miniature EPSCs.* AMPAR-mediated mEPSCs were recorded from individual L5 pyramidal neurons in the anterior cingulated (ACC) cortex and the prelimbic (PrL) cortex. Pyramidal neurons were identified morphologically by infrared differential interference contrast microscopy. Whole-cell recordings were performed under voltage-clamp mode (holding at −70 mV) in the presence of 1 μM tetrodotoxin (TTX; Sigma) in ACSF. Series resistance was monitored throughout the recordings and data were discarded if the resistance changed by more than 15%. Signals were filtered offline at 2 kHz and analyzed with Mini Analysis 6 (Synaptosoft). Average mEPSC amplitude and frequency, as well as cumulative probability for mEPSC

amplitude and inter-event interval for each cell, was calculated by collecting mEPSCs recorded during the initial 5-min period after whole-cell access and stable series resistance was achieved.

**Behavior.** For Nsun2 overexpression studies, four separate cohorts of mice were used for behavioral experiments and testing commenced in order as follows: Cohort 1: Open field, Light/dark box, Forced Swim, Tail Suspension. Cohort 2: Elevated Plus Maze. Cohort 3: Y-maze Cohort 4: Fear conditioning. For Nsun2 knockout studies, 2–3 cohorts of mice were used for anxiety- and behavioral despair tests and all experienced testing in order as follows: Open field, Light/dark box, Elevated Plus Maze, Tail Suspension, Forced Swim. In addition, two separate cohorts of mice were used for only fear conditioning studies and one separate cohort was used for Y-maze. All behavioral testing was conducted during the animals' light cycle and took place ~21 days after viral injection for Nsun2 over-expression studies or at an identical age in Nsun2 KO and WT mice (13–16 weeks of age). For all behavioral tests, mice were allowed to habituate in the testing room for 30 min prior to testing, and testing was conducted in normal fluorescent room light.

*Open field.* The open-field chamber consisted of a white Plexiglas box (40 × 40 × 30 cm high). Mice were placed individually into the box for 20 min. Locomotor activity and time spent in an imaginary center square (15 × 15 cm) of the open field were recorded using the Fusion 5.0 Superflex system.

*Light/dark box.* The light–dark box consisted of the open field chamber (described above) with a black insert dividing the arena into dark and light components. Mice were placed in the dark compartment and able to explore freely between both halves of the chamber through a small opening in the divider for 10 min. The activity was recorded with Fusion 5.0 Superflex system. Latency to enter the light compartment, duration of time spent in the dark and light chambers, and distance traveled in the chambers were measured.

*Elevated plus maze.* The apparatus consisted of a plus-shaped maze elevated 65 cm off the ground with two arms containing high black walls and two open arms with no walls. At the start of testing, mice were placed in the center of the maze (5 × 5 cm) and allowed to freely explore for 8 min. We measured time spend in the open arms vs. closed arms and total locomotor activity using Ethovision (v.14) software.

*Tail suspension.* Mice were suspended in mid-air by their tail using laboratory tape and allowed to move freely for 5 min. Video recordings of each session were recorded and total time immobile was measured. After each session, mice were immediately placed back in the home cage. All mice in each cage were tested simultaneously.

*Forced swim.* Mice were placed in a transparent glass 5-liter beaker filled with ~3 litems room temperature tap water (~23–25°C) and allowed to swim freely for 5 min. Video recordings of each session were recorded and total time immobile was measured. After each session, mice were immediately dried of excess water and placed back in the home cage. All mice in each cage were tested simultaneously.

*Y-maze spontaneous alteration.* Mice were gently handled for 2 days prior to testing and habituated to the testing room for 1 h the day of testing. Mice were placed in one arm of a traditional Y-maze and allowed to roam freely for 8 min. Video recordings of each session were recorded with an overhead camera, and the order of entries into arms was scored by hand. The total number of alternations was divided by total arm visits to achieve an alternation percentage score.

*Contextual and cues fear conditioning.* Mice were gently handled for 4 days and habituated to the fear-conditioning room for 2 days prior to training and testing. All fear conditioning took place in Med Associates conditioning chambers (VFC-008) and freezing was assessed using Med Associates Video Freeze Software (Med Associates, St Albans, Vt). Animals were initially administered three tone-shock pairings (20 s, 75 dB white noise co-terminating with a 2 s, 0.5 mA shock). A 3-min. baseline period preceded the first tone, and there were 2 min. between each tone shock pairing, as well as after the final pairing before being taken out of the chamber. Initial conditioning was performed in a chamber with distinct lighting (500 Lux white light), flooring (grid floor), and scent (5% simple green solution). The day after conditioning, animals were placed back into this chamber in order to assess contextual fear during a 10-min. test. In order to assess fear of the tone in the absence of pre-tone freezing differences, animals were placed in an alternative conditioning chamber over the course of 5 days, 20 min. each (dark chamber, solid acrylic floor, 1% acetic acid scent). On a final 6th day, at a point where there were no group differences in baseline freezing in the chamber, animals were exposed to the tone three times using the same time intervals used during initial training.

**Statistical analyses.** Statistical analyses were performed using Graphpad Prism 8.4.3 software. For western blot results quantifying protein expression of Nsun2 and RT-qPCR results quantifying *Nsun2* gene expression, unpaired $t$ tests were

used to compare genotypes or treatment groups. For electrophysiology experiments, average mEPSC amplitude and frequency were compared between WT and KO by unpaired two-tailed $t$ tests. For behavioral tests, two-way ANOVAs (genotype or AAV injection × sex) and two-tailed $t$ test with Bonferroni correction as posthoc tests were used when appropriate. For fear conditioning, we used two-way repeated-measures ANOVAs (genotype × CS presentation or context) and two-tailed t-tests with Bonferroni correction as post hoc tests when appropriate. All unpaired $t$ tests were two-tailed except for Neurogranin protein expression in synaptosomes (Fig. 4b), which was one-tailed due to the confirmatory nature of the experiment based on the previous proteomics (LC-MS/MS) result for that protein. Simple linear regression was performed to calculate the relationship between two variables (and if the slope was significantly non-zero) for protein vs. RNA expression and Pearson correlation was used for codon content vs. logTER. Statistical significance was denoted by $P < 0.05$. All bar graphs are presented as the mean with error bars representing the standard error of the mean (SEM) and include all individual data points.

**Reporting summary**. Further information on research design is available in the Nature Research Reporting Summary linked to this article.

## Data availability

The raw and processed mouse RNA, tRNA, YAMAT, and Ribosome sequencing data generated in this study have been deposited in the Gene Expression Omnibus database under accession code GSE165202. The mass spectrometry proteomics data used in this study are available through the ProteomeXchange Consortium via the PRIDE[72] partner repository under accession code PXD023437. All other data generated in this study are provided in the Supplementary Information and Source Data file. Source data are provided with this paper.

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

## Acknowledgements

We thank members of the Akbarian lab, especially Dr. Lucy Bicks and Dr. Sandhya Chandrasekaran for discussion and helpful suggestions. We thank Dr. Andrew Chess for providing sequencing equipment and Dr. Ian Maze and his laboratory for western blotting expertise. This work was supported by NIH postdoctoral fellowship F32MH115565-01A1 (J.B.) and NIH grants P50MH096890 (E.J.N., S.A.), R01MH117790-01 (S.A.), R01MH104341 (S.A.), R01MH106489 (W.D.Y.), and by JJPVAMC-CSR&D 1I01CX001395-01 (F.H.). This work was also partially supported by NIH grants DP2MH122399 (D.J.C.), 1R01MH120162 (D.J.C.), and a DoD PRCRP Horizon Award W81XWH-19-1-0594 (A.N.).

## Author contributions

J.B., S.A., and F.H. conceptualized and designed experiments, interpreted the data, and wrote the manuscript with input from T.P., H.G., C.G.H., and W.D.Y. J.B. collected all mouse brain tissue, collected and analyzed the data for tRNA expression/methylation, RNA/protein abundance, and behavioral tests. C.P.W. built DM-tRNA-seq libraries, and M.F. and C.D.K. analyzed DM-tRNA-seq data. S.H. and H.M. performed LC-MS/MS and analyzed proteomics data. B.R., H.Al., and H.M. performed metabolomics assays for amino acids and analyzed the data. A.N. and H.G. created YAMATseq/UMI protocol and analysis pipeline and performed RiboSeq experiments. A.N., S.M., H.As., and H.G. analyzed RiboSeq data. H.G. provided translation efficiency and codon content analysis. A.P.J. ran tRNA libraries on MiSeq. B.J. performed immunohistochemistry staining and microscopy. S.E.G. analyzed RNAseq data. H.L.P. and W.D.Y. conducted electrophysiological recordings and analyzed the data. J.B., Z.T.P., and D.J.C. acquired, analyzed, and interpreted fear conditioning data.

## Competing interests

The authors declare no competing interests.
