## [Peer Review File · Nature Communications]

Reviewers' Comments:

Reviewer #1:

Remarks to the Author:

In this manuscript, Blaze et al characterize the role nsun2 in neurons in a mouse knockout model. Blaze et al report that nsun2 targets (among others) tRNA-Gly, that levels of tRNA-Gly are reduced in the knockouts, causing a specific reduction in translation and translational efficiency of Gly containing proteins. Blaze et al then characterize the phenotypes of tRNA-Gly depletion on synaptic signalling and behavior.

Generally, we find this manuscript to be well-written, informative, and compelling. We appreciate the rigorous controls conducted by the authors, for example in quantification of tRNA levels by three different approaches.

One aspect that might need a better clarification in the text pertains to defining the boundaries between the novel findings in this manuscript, versus ones that are known. The fact that tRNA-Gly is a target of nsun2 is well-established. Also the fact that tRNA-Gly stability is nsun2 dependent has been characterized by studies by the Suzuki and Frye groups. In our view there is importance in replicating these findings, and in performing them in neurons, but precisely (textually) defining the parts that replicate previous findings versus ones that go beyond is in our view important.

One aspect that I do believe the authors are establishing, for the first time, is the impact on translation of tRNA-Gly harboring proteins. The authors establish this by comparing mass-spectrometry data to RNA-seq. Given that this is, in my view, one of the central findings in this paper, I would strongly encourage the authors to apply ribo-seq for this question. Ribo-seq is not only the state of the art for this kind of analysis, but would also provide an ability to analyze the data at codon-resolution, and to assess whether ribosomes are stalled at tRNA-Gly codons, which would strongly support their results.

Given our lack of expertise in neurology, we will not comment on the last section of the manuscript in which various nsun2-mediated phenotypes are documented.

Finally, we have the three minor comments below: t

1. Legends should clearly explain the figures, for example in figure 1b, the scheme is not on the right as stated, in extended figure 2f, the panels are not top and bottom to each other.
2. Also in figure 1C, the circles in one of the darker colors (green and blue) should be in a lighter shade, since when printing the figure or the legend in grey scale the colors of the circles seem identical.
3. We found figure 1D a bit confusing. The combination of colors, images, and indicators of significance all combine into a somewhat unusual plot that is difficult to interpret.

Reviewer #2:

Remarks to the Author:

Loss of function of NSUN2 in mouse and human led to motor, neurodevelopmental and cognitive defects, suggesting the neurological function of 5mC in RNA (Abbasi-Moheb et al. 2012; Khan et al. 2012; Martinez et al. 2012; Fahiminiya et al. 2014). As the authors have correctly pointed out, the biological role of this RNA methyltransferase in regulating "proteome" in the nervous system was unknown. This study is timely and important, but the current results and the description/interpretation of the results are insufficient for explaining the molecular changes or the behavioral deficits associated with the molecular changes. Substantial analysis and interpretation are required for completing this study.

In this article, Blaze et al used an NSUN2 forebrain neuronal depletion mouse model (KO, Nsun2 2lox/2lox crossed with "CamK-Cre"), and an NSUN2 prefrontal cortex neuronal overexpression model (OE, by injecting AAV8 virus carrying hSyn1-Nsun2-GFP), to investigate how alterations in NSUN2 protein expression level may impact the animals at molecular, cellular, and behavioral

levels.

The authors first showed methylation sites on tRNAGly GCC, tRNAGlu TTC, and tRNAAsp GTC that NSUN2 was known to be responsible for became hypomethylated in the NSUN2 cKO brain tissue, and some sites became hypermethylated upon NSUN2 overexpression. Then the authors used multiple tRNA/RNA/tRF sequencing techniques to quantify the expression changes in tRNA isodecoders and found significant downregulation of 7 Glycine tRNA isodecoders. Next, the authors used LC-MS/MS to quantify protein expression changes in the KO animals and found that the proteins whose expression was downregulated had on average a higher glycine content (although the difference seemed quite small, possibly less than 1%) and negatively correlated to the glycine content. In addition to molecular characterization of the KO mice, the authors measured mEPSCs in ACC and PrL cortex and found decreased frequency and unaltered amplitude in ACC, while unaltered frequency but decreased amplitude in PrL. Next, the authors demonstrated that the KO mice had impaired contextual fear memory and showed antidepressant-like behaviors, opposite from the OE mice. Finally, the authors showed that an injection of antidepressant ketamine, caused significant decrease of methylation level on C46, C47, and C48 of tRNAGly GCC in 24 hours.

Major points:

1. Neither of the two claims in the abstract finds sufficient supporting evidence. Behavioral alteration by overexpressing NSUN2 in PFC resulted in increased immobile time in forced swimming test and tail suspension test in males (but not in females), but not in open field test, or light/dark transition test, or elevated plus maze. There is no memory test conducted on the OE mice. On the other hand, the knockout males showed contextual fear memory deficits and increased time spent in open arm in elevated plus maze, but no cued fear memory deficits; and the knockout was not specific to PFC. Thus "fear memory and antidepressant-like....highly sensitive to bidirectional changes....in PFC neurons" (line 38) hardly matches the results authors reported. The second claim was as well unsupported. Table 1 pointed to 12 tRNA with significantly altered methylation level thus "a selective deficit in multiple glycine tRNAs" (line 40) is unsupported by the results. "deficits in glycine-rich neuronal proteins" hardly capture the results with hundreds of proteins with their expressions either up- or down-regulated (Fig 2b).
2. The length of the main text describing the authors' findings was 112 lines. The information was insufficient for describing/interpreting the results. For example, Line 110-118 described the results of electrophysiological recordings with no explanation to why the two regions were selected instead of PFC, the target region where NSUN2 was overexpressed. Moreover, the results in ACC and PrL were inconsistent with each other. What is the explanation? Are synaptic proteome composition differentially affected in these two regions?
3. Is it possible that the downregulation in the proteome was due to overall repression of proteins in excitatory neurons instead of restricted repression of glycine-rich proteins? Since the proteome analysis is done on tissue lysate of a mixture of all cell types including neurons, functional categories related to excitatory neuron function such as "glutamatergic synapse" can become enriched when the repression occurs in KO neurons.
4. The characterization of NSUN2-cKO mice is unclear. This reviewer is not sure what CamK-Cre line the authors were referring to. Assuming the authors meant CamK2 α -cre mice lines, which has multiple lines with different Cre expression patterns, it is important to know which specific line was used in this study. For details, please see [https://doi.org/10.1016/S0092-8674\(00\)81826-7](https://doi.org/10.1016/S0092-8674(00)81826-7). Without further information and description, the identity of the cells where KO is occurring in the mice is unclear.
5. What is the general health of the cKO mice like?
6. In Extended figure 2b, 5'tRF was also decreased, as well as the mature tRNA. Thus an instability-related mal-production of tRF can not explain the decrease of specific tRNAs. What is authors explanation about the distinct results in embryonic brain (Blanco S et al., 2014)?

7. No potential functional links between the proteome alteration or the e-phys results and behavior were provided or attempted.

8. NSUN2 targets a variety of tRNAs including glycine isodecoders. The quantitative results on other known NSUN2 target tRNAs need to be carefully investigated and described (doi: 10.3390/genes10020102).

9. In addition to tRNA, NSUN2 also targets vault RNA, an abundant lncRNA, and some mRNAs in neurons. The impact to these RNAs in the KO and OE mice should be described and explained (doi: 10.3390/genes10020102).

10. Authors claimed "ketamine...significant decrease in tRNAGlyGCC methylation 24 hours post-injection" (Line 144-145), but no actual values were mentioned in the main text. The quantification graph in Fig 3g and data in Supplementary table 9 showed highly similar methylation level between saline and ketamine treated animals. If there is a reduction, the reduction is in a range of 0.0-0.3%. Is this level of reduction sufficient for linking the anti-depressant effect of ketamine and behavior in KO mice? This reviewer can not comfortably conclude the same with the authors.

11. Line 135-139. The conclusion on translational-control of glycine-rich proteins as the 3rd glycinergic regulatory pathway of central nervous system is not convincing.

12. There is no "Discussion" section.

13. To address the role of NSUN2 in PFC, a functional rescue using AAV8-Nsun2-GFP in KO background is preferred.

Minor points:

1. Please provide primer sequences used for RT-PCR in this study.

2. What does the Y-axis in Fig 1a represent for the "RNA-seq" profile? At what age were the mice sampled?

3. In Fig 1b, no merged image is provided to show neuron-expression of NSUN2 in the cortex.

4. The authors described Fig 1c as "heatmaps", but they are not by definition. They are tables representing methylation status on each read. The number of rows should be described in the legend.

5. There is no sufficient data suggesting Fig 3b

6. I could not find information to describe the ages of the mice used for each experiment.

7. P value as 0.06 (Fig 3) was considered significant in this study. This is not conventional, please provide rationale for "significant".

8. In Fig 2a and Extended Fig 2a, statistical analysis results are not depicted.

Reviewer #3:

Remarks to the Author:

In their manuscript Blaze and colleagues demonstrate that NSUN2-mediated tRNA methylation is required for the efficient translation of glycine-rich mRNAs in the postnatal mouse brain. The codon-specific shift of translation affects the global neural proteome and alters synaptic signalling and behaviour.

tRNA is the most extensively modified RNA, and many of these modifications have been linked to

human diseases including neurological disorders. Mutations in the tRNA methyltransferase NSUN2 has been widely linked to neurological deficits in mouse and human. However, all previous studies focused on the role of NSUN2-specific tRNA methylation during development. Hence, the functional importance of NSUN2 in the postnatal brain is currently unknown. Moreover, the precise underlying molecular mechanisms of how NSUN2-mediated methylation controls mRNA translation rates are unclear. Here, the authors show for the first time that NSUN2-specific methylation of tRNAs determines codon usage. Given that dysfunctional RNA processing and translation in neuronal tissue is often observed in neurological diseases, this study is a timely and important contribution to the field. Overall, this is a very mature manuscript but is very dense, which makes it at times difficult to follow and the novel findings are at times a little lost.

1. Page 2; line 49: Only when reading the methods section, it becomes clear that the generated lines deplete NSUN2 postnatally using the CamK-Cre mice. This is very different from published mouse lines where only the developing brain was analysed. The function of NSUN2 in the postnatal brain is entirely unknown. Moreover, over-expression experiments have also not been performed before. I recommend highlighting the two facts better and early on in the text.

2. Page 3; line 83: Can the authors comment on why glycine is specifically affected? Could it be that this is simply due to the fact that in the brain all 3 cytosines in the variable loop are methylated by NSUN2. So, tRNAGly would be hypermethylated when compared to other tissues?

3. Page 4; line 92: Are neural proteins in general enriched in Gly when compared to all proteins?

4. I believe figure 3b is not referred to in the text.

5. If glycine-rich mRNAs are less translated, does this cause an increase of the amino acid glycine in the cells?

Dear Reviewers,

We very much appreciate the highly constructive comments of each of the three Reviewers and provide below a detailed point-by-point response, and an overview table summarizing the newly added experiments and data analyses. We are pleased to report that we were able to address each of the Reviewers' comments.

Reviewer request	Reviewer requested	New experiments/analysis	Figure	Result
Ribosome profiling	1	Ribosome profiling (RiboSeq)	Fig. 2e; Extended Data Fig. 5b-d	Slower decoding (increased ribosome dwell time) for the GGA codon, which corresponds to tRNA GlyTCC. Also, a significant association between translation of GGA codons and logTERs using RiboSeq data.
Fear conditioning after Nsun2 overexpression	2	Fear conditioning in AAV-Nsun2 vs. AAV-GFP mice	Extended Data Fig. 6d; Suppl. Table 12	No effect of PFC Nsun2 overexpression on fear memory acquisition, contextual fear memory, or cued fear memory
		Y-maze in AAV-Nsun2 vs. AAV-GFP for working memory	Extended Data Fig. 6d; Suppl. Table 12	No effect of PFC Nsun2 overexpression on Y-maze working memory performance
Mechanistic link between PFC Nsun2 and bidirectional changes in behavior	2	PFC-specific Nsun2 KO using AAVCre injection	Extended Data Fig. 1e; Suppl. Table 13	AAVCre injection into adult PFC to knockdown Nsun2 in adult cortex decreases Nsun2 expression and decreases GlyGCC tRNA methylation
		Behavioral despair tests in AAV-Cre Nsun2 KO vs. AAV-GFP mice	Fig. 5e, Suppl. Table 12	AAVCre injection for PFC-specific Nsun2 KO decreased depressive-like behavior in the tail suspension test.
Cytosine methylation data for additional tRNAs	2	Bisulfite amplicon sequencing for additional tRNAs;	Extended Data Fig. 2a	ProTGG and ValAAC show decreases in methylation for Nsun2 KO and no change after Nsun2 OE.
General health of Nsun2 KO mice	2	Brain and body weight for Nsun2 KO mice	Extended Data Fig. 1d; Suppl. Table 1	Decreased brain and body weight for Nsun2 KO mice compared to WT mice
		Mendelian ratios of Nsun2 KO offspring	Extended Data Fig. 1d	Observed Mendelian ratios from breeding not significantly different from expected ratios
Mechanistic link between ephys phenotype and tRNA/proteomics	2	Identification of downregulated proteins in Nsun2 KO that are involved in synaptic transmission	Extended Data Fig. 4f; Suppl. Table 7	In our set of downregulated proteins, we identified 50 involved in synaptic transmission, which are even more enriched for glycine than the total downregulated population.
Nsun2 staining in neurons	2	In situ hybridization for Nsun2 exon 6 expression	Extended Data Fig. 1b	Detected a decrease in Nsun2 exon 6 mRNA in the KO cortex.
Glycine amino acid measurement	2,3	Identification of significantly altered proteins involved in glycine biosynthesis	Fig. 3a; Suppl. Table 10	Four glycine biosynthesis-related proteins significantly upregulated after Nsun2 KO and 1 downregulated
		Unbiased metabolomics screen in Nsun2 KO	Fig. 3b; Suppl. Table 11	Of 19 amino acids, glycine is increased in Nsun2 KO while no other amino acids show significant alterations
Explanation for glycine-selective deficit in tRNAs after widespread m5C loss	3	Analysis of data from Blanco et al. (2014) to identify mouse tRNAs with most Nsun2-methylation sites	Extended Data Fig. 3b	In mouse skin cells, three tRNA families have 3 Nsun2-methylated cytosines (Gly, Glu, Pro)
		Comparison of wild-type cortex cytosine methylation in 3 tRNA with most Nsun2-methylation sites	Extended Data Fig. 3c	In our cortex methylation data of the three (Gly, Glu, Pro) tRNAs that have most Nsun2-methylated sites, we identify Gly tRNAs as having the most Nsun2-methylated sites in brain (4 sites) in addition to a DNMT2 site not present in other tissue types.

ProteomeXchange Consortium login info for Reviewers:

Username: reviewer_pxd023437@ebi.ac.uk

Password: bXsz3XWn

Reviewer #1:

In this manuscript, Blaze et al characterize the role nsun2 in neurons in a mouse knockout model. Blaze et al report that nsun2 targets (among others) tRNA-Gly, that levels of tRNA-Gly are reduced in the knockouts, causing a specific reduction in translation and translational efficiency of Gly containing proteins. Blaze et al then characterize the phenotypes of tRNA-Gly depletion on synaptic signalling and behavior.

Generally, we find this manuscript to be well-written, informative, and compelling. We appreciate the rigorous controls conducted by the authors, for example in quantification of tRNA levels by three different approaches.

1. *One aspect that might need a better clarification in the text pertains to defining the boundaries between the novel findings in this manuscript, versus ones that are known. The fact that tRNA-Gly is a target of nsun2 is well-established. Also the fact that tRNA-Gly stability is nsun2 dependent has been characterized by studies by the Suzuki and Frye groups. In our view there is importance in replicating these findings, and in performing them in neurons, but precisely (textually) defining the parts that replicate previous findings versus ones that go beyond is in our view important.*

Response: We appreciate this comment and in response:

(i) added a comprehensive introductory paragraph (Line 66-76) describing the previous work on Nsun2-induced tRNA methylation:

Evidence for the importance of solely NSUN2-mediated m⁵C in human health and disease was first provided by individuals with loss-of-function mutation in NSUN2, displaying intellectual disability (ID), facial dysmorphism and distal myopathy¹⁰⁻¹³. The idea that NSUN2 deficiency causes neurological abnormalities was further delineated by studies in Drosophila (d) and mouse^{9,10}, including dNsun2 knockdown with impaired short-term memory after aversive olfactory conditioning that was rescued by pan-neuronal expression of dNsun2¹⁰. Furthermore, mice with Nsun2 germline deletion demonstrated various impairments in locomotor activity and behavior together with reduced brain size due to excessive cell death in prenatal brain⁹. The mechanism thus far suggested for these deficits is impaired translation induced by increased tRNA fragmentation after Nsun2 ablation^{14,15}. However, although tRNA cytosine methylation (m⁵C) has recently been profiled in various tissues including embryonic brain^{9,16-22} the m⁵C RNA methylome and its functional relevance in the mature adult mammalian brain is still unexplored.

(ii) We then added (Line 76-88) added a comprehensive paragraph describing how our work differs from previous studies to show new functions of Nsun2 in adult brain function:

In the current study, we use three different genetic approaches selectively targeting NSUN2 function in differentiated neurons of the postnatal and adult brain. These include neuron-specific conditional Nsun2 ablations and transgene-mediated increase in Nsun2 expression and methylation activity in adult prefrontal cortex (PFC), thereby focusing for the first time on the effect of NSun2 enzymatic activity on specific neuronal subpopulations in the mature brain. We report that Nsun2 shapes complex behaviors with neuronal function highly sensitive to bi-directional changes in tRNA methyltransferase activity. We show that the underlying mechanisms include alterations in tRNAs defined by high cytosine methylation specifically in the variable loop region, resulting in prominent deficits of glycine-specific tRNA isodecoders with corresponding shifts in the neuronal proteome due to decreased translational efficiency of glycine-rich neuronal proteins. Ultimately, these distortions in the glycinergic neuronal translome lead to a striking 2.46-fold increase in PFC glycine levels associated with multi-fold increases in several key enzymes of the glycine biosynthetic pathway, further illustrating that in the adult brain, proteomic and metabolic homeostasis associated with specific amino acids is linked to epitranscriptomic regulation of cognate tRNAs.

2. *One aspect that I do believe the authors are establishing, for the first time, is the impact on translation of tRNA-Gly harboring proteins. The authors establish this by comparing mass-spectrometry data to RNA-seq. Given that this is, in my view, one of the central findings in this paper, I would strongly encourage the authors to apply ribo-seq for this question. Ribo-seq is not only the state of the art for this kind of analysis, but would also provide an ability to analyze the data at codon-resolution, and to assess whether ribosomes are stalled at tRNA-Gly codons, which would strongly support their results.*

Response: We thank the Reviewer for this suggestion and have now conducted an additional experiment, performing RiboSeq on n=2 KO/2 WT cortex samples. Results show a dramatic increase in ribosome dwell time for GGA codons (corresponding to TCC anticodon for Gly compared to other Gly codons and other non-Gly codons, which corresponds to a drastic decrease in GlyTCC tRNA expression (Fig. 2a). We have added this RiboSeq data as an additional panel in Fig. 2e and in Extended Data Fig. 5b-d. Additionally, we used LogTERs from our RiboSeq data and identified a strong association between GGA codon-enrichment in transcripts and low LogTER mRNAs. We have presented this data in the Results section line 170-184.

Given our lack of expertise in neurology, we will not comment on the last section of the manuscript in which various nsun2-mediated phenotypes are documented.

Finally, we have the three minor comments below:

1. *Legends should clearly explain the figures, for example in figure 1b, the scheme is not on the right as stated, in extended figure 2f, the panels are not top and bottom to each other.*

Response: We have adjusted the legends to reflect the figures and appreciate the Reviewer's attention to this detail.

2. Also, in figure 1C, the circles in one of the darker colors (green and blue) should be in a lighter shade, since when printing the figure or the legend in grey scale the colors of the circles seem identical.

Response: We have also changed the color of the blue and green circles in Fig. 1 and Extended Data Fig. 3 to a lighter shade as requested by the Reviewer.

3. We found figure 1D a bit confusing. The combination of colors, images, and indicators of significance all combine into a somewhat unusual plot that is difficult to interpret.'

Response: We agree that Figure 1D has a lot of data and may be difficult for the reader to interpret, so we have made changes to make this figure clearer. First, we have significantly enlarged the figure panel and included lines to separate the 2 conditions (KO and AAV Nsun2 overexpression), each replicate (2 biological replicates each), and each cytosine. We have also added more description in the heat map legend denoting it represents logFC of the methylation percentage for each subject relative to the proper control.

Reviewer #2:

Loss of function of NSUN2 in mouse and human led to motor, neurodevelopmental and cognitive defects, suggesting the neurological function of 5mC in RNA (Abbasi-Moheb et al. 2012; Khan et al. 2012; Martinez et al. 2012; Fahiminiya et al. 2014). As the authors have correctly pointed out, the biological role of this RNA methyltransferase in regulating ???proteome??? in the nervous system was unknown. This study is timely and important, but the current results and the description/interpretation of the results are insufficient for explaining the molecular changes or the behavioral deficits associated with the molecular changes. Substantial analysis and interpretation are required for completing this study.

In this article, Blaze et al used an NSUN2 forebrain neuronal depletion mouse model (KO, Nsun2 2lox/2lox crossed with ???CamK-Cre???), and an NSUN2 prefrontal cortex neuronal overexpression model (OE, by injecting AAV8 virus carrying hSyn1-Nsun2-GFP), to investigate how alterations in NSUN2 protein expression level may impact the animals at molecular, cellular, and behavioral levels.

The authors first showed methylation sites on tRNAGly GCC, tRNAGlu TTC, and tRNAAsp GTC that NSUN2 was known to be responsible for became hypomethylated in the NSUN2 cKO brain tissue, and some sites became hypermethylated upon NSUN2 overexpression. Then the authors used multiple tRNA/RNA/tRF sequencing techniques to quantify the expression changes in tRNA isodecoders and found significant downregulation of 7 Glycine tRNA isodecoders. Next, the authors used LC-MS/MS to quantify protein expression changes in the KO animals and found that the proteins whose expression was downregulated had on average a higher glycine content (although the difference seemed quite small, possibly less than 1%) and negatively correlated to the glycine content. In addition to molecular characterization of the KO mice, the authors measured mEPSCs in ACC and PrL cortex and found decreased frequency and unaltered amplitude in ACC, while unaltered frequency but decreased amplitude in PrL. Next, the authors demonstrated that the KO mice had impaired contextual fear memory and showed antidepressant-like behaviors, opposite from the OE mice. Finally, the authors showed that an injection of antidepressant ketamine, caused significant decrease of methylation level on C46, C47, and C48 of tRNAGly GCC in 24 hours.

Response: We agree with the Reviewer that the manuscript would benefit from a substantial increase in description and interpretation. We have now:

(i) elongated the manuscript with a more thorough Introduction (newly added text includes all of lines 60-88) and Discussion (newly added text includes all of lines 246-323). This newly added text in the discussion starts with a lead-in sentence 'Which molecular mechanisms link Nsun2 to these robust behavioral and physiological phenotypes?' Lines 252-278 connect high Gly tRNA methylation to the fact that these tRNAs are selectively decreased after Nsun2 knock-out in mature neurons, then draw a link the decreased translational efficiencies of Gly-rich proteins including synaptic proteins, thereby drawing a link to the electrophysiological deficits and

behavioral changes. To more directly address the Reviewer's comments, we now also included a new Table (Suppl. Table 7) and Figure (Extended Data Fig. 4f) listing 50 key synaptic proteins that are downregulated in mutant, for which we now show data for glycine composition. Extended Data Fig. 4f demonstrates that these 50 key synaptic proteins (downregulated in Nsun2 mutant) show an even higher mean glycine percentage. In addition, we conducted for this resubmission additional metabolomic studies, showing a dramatic 146% increase in cortical glycine level, an alteration that was highly specific to this particular canonical amino acid. We now discuss the molecular alterations in the glycine biosynthesis pathway in our mutant mice in lines 279-304.

Major Points

1. *Neither of the two claims in the abstract finds sufficient supporting evidence. Behavioral alteration by overexpressing NSUN2 in PFC resulted in increased immobile time in forced swimming test and tail suspension test in males (but not in females), but not in open field test, or light/dark transition test, or elevated plus maze. There is no memory test conducted on the OE mice. On the other hand, the knockout males showed contextual fear memory deficits and increased time spent in open arm in elevated plus maze, but no cued fear memory deficits; and the knockout was not specific to PFC. Thus fear memory and antidepressant-like highly sensitive to bidirectional changes in PFC neurons (line 38) hardly matches the results authors reported. The second claim was as well unsupported. Table 1 pointed to 12 tRNA with significantly altered methylation level thus a selective deficit in multiple glycine tRNAs (line 40) is unsupported by the results. deficits in glycine-rich neuronal proteins hardly capture the results with hundreds of proteins with their expressions either up- or down-regulated (Fig 2b).*

Response: We thank the Reviewer for pointing this out. As requested by the Reviewer, we have now thoroughly revised the abstract and also:

- (i) conducted two additional experiments assessing fear memory and Y-maze working memory tests on Nsun2 overexpressing mice vs. controls. These additional data are shown in the newly added Extended Data Fig. 6d and in the text Lines 232-233.
- (ii) included a new genetic manipulation of Nsun2 to produce a PFC-specific knockdown of Nsun2, for which we 1) confirmed a significant decrease in *Nsun2* expression and GlyGCC cytosine methylation and 2) performed tests of behavioral despair and confirmed an anti-depressant phenotype similar to the Camk2a-Cre forebrain Nsun2 KO. The validation of this third genetic manipulation is in Extended Data Fig. 1e and Suppl. Table 13, and behavioral data is presented in Fig. 5e and Suppl. Table 12.

For our second claim, we have clarified our reference to Suppl. Table 2 at line 99 that this refers to “tRNA expression levels independent of epitranscriptomic modification” in the text to make it clearer to the reader. Table 1 points to 162 tRNA isodecoders of which 12 tRNAs with significantly altered expression levels, with 7 of those being Gly tRNA isodecoders, which we have validated with two other techniques (YAMAT-seq and qPCR).

While hundreds of proteins were up and downregulate, a strength of our study was the advanced bioinformatic analysis using the FIRE heatmap (Fig. 2d) to correlate glycine codon content with translation efficiencies. When added to our previous analysis of amino acid content of proteins downregulated vs. upregulated (Fig. 2c), the data shown in Fig.2d further strengthens the link between deficits in Gly-rich tRNAs in the cell and the downregulation of glycine-rich proteins. To draw a more precise parallel between synaptic functioning and our proteomic outcomes, (as discussed in comment 1) we have also included a new Table (Suppl. Table 7) and Figure (Extended Data Fig. 4f) listing 50 key synaptic proteins that are downregulated in mutant, for which we now show data for glycine composition. Extended Data Fig. 4f demonstrates that these 50 key synaptic proteins (downregulated in Nsun2 mutant) show an even higher mean glycine percentage. We have also revised the abstract to reflect these findings.

2. *The length of the main text describing the authors findings was 112 lines. The information was insufficient for describing/interpreting the results. For example, Line 110-118 described the results of electrophysiological recordings with no explanation to why the two regions were selected instead of PFC,*

the target region where NSUN2 was overexpressed. Moreover, the results in ACC and PrL were inconsistent with each other. What is the explanation? Are synaptic proteome composition differentially affected in these two regions?

Response: We agree with the Reviewer that explanation of our findings requires a significant increase in length of the manuscript and have substantially elongated the text and descriptions of experiments.

(i) In response to the reason for choosing ACC and PrL for electrophysiological experiments, we now specify in the results section that:

“...we performed whole cell patch clamp recordings on individual layer V pyramidal neurons in the mouse PFC to measure AMPA-receptor mediated miniature(m) EPSCs in adult *Nsun2* mutants and controls. We specifically chose two sub-regions of the rodent PFC, the anterior cingulate cortex (ACC) and prelimbic cortex (PrL), which have been repeatedly implicated in rodent cognition (ref) and depressive-like behavior (ref)”

(ii) We also discuss in lines 321-323 in the Discussion that investigating proteomic changes induced by tRNA alterations in the distinct PFC subregions as a future direction necessary for characterizing impairments to PFC glutamatergic signaling.

3. *Is it possible that the downregulation in the proteome was due to overall repression of proteins in excitatory neurons instead of restricted repression of glycine-rich proteins? Since the proteome analysis is done on tissue lysate of a mixture of all cell types including neurons, functional categories related to excitatory neuron function such as glutamatergic synapse can become enriched when the repression occurs in KO neurons.*

Response: We agree with the Reviewer that because our *Nsun2* KO was specific to *Camk2a*-expressing excitatory glutamatergic neurons, it is likely that the downregulated proteins are in that cell type, and we have now noted in the discussion that a limitation of our proteomic experiments is the lack of cell specificity due to input restrictions. We have now added an additional analysis to our proteomics data to attempt to answer this question. To identify whether the repression of glycine-rich proteins was due to the neuronal proteome, and further glutamatergic neuron proteins, being enriched for glycine, we conducted additional analyses and have specified in the Results section lines 151-156 the following text as well as adding the new data to new Extended Data Fig. 4f:

“These alterations were highly specific because glycine content of the entire set of n=434 neuron-specific proteins (UniProtKB) in our cortical proteome (comprised of neuronal and glial proteins) was very similar to the glycine content of the total proteome in our cortex homogenates (Mann-Whitney test, p=0.412; Extended Data Fig. 4f), confirming that the observed enrichment of high glycine content proteins in the fraction of down-regulated proteins in the *Nsun2* mutant cortex do not reflect a non-specific decline of the neuronal proteome overall. ”

4. *The characterization of NSUN2-cKO mice is unclear. This reviewer is not sure what CamK-Cre line the authors were referring to. Assuming the authors meant CamK2a-cre mice lines, which has multiple lines with different Cre expression patterns, it is important to know which specific line was used in this study. For details, please see [https://doi.org/10.1016/S0092-8674\(00\)81826-7](https://doi.org/10.1016/S0092-8674(00)81826-7). Without further information and description, the identity of the cells where KO is occurring in the mice is unclear.*

Response: We now specify in the Methods section the timing of the KO using the *Camk2a-Cre* line and cite previous papers from our lab using the same line:

“*Nsun2*^{2lox/2lox} mice were crossed with a *CamK-Cre*⁺ line to produce *CamK-Cre*⁺,*Nsun2*^{2lox/2lox} mice for knockout of *Nsun2* in excitatory forebrain neurons. The calmodulin-kinase II (*CamK*)-*Cre* transgenic line results in widespread neuronal *Cre*-mediated deletion in forebrain before postnatal day 18 as previously described (refs)”

5. *What is the general health of the cKO mice like?*

Response: We have added new data in the text (lines 92-96) about the general health of the cKO mice, including:

(i) size of conditional CK-Cre KO mice showing significant decrease in body and brain weight of Nsun2 KO mice.

(ii) that we observed the expected Mendelian ratios of offspring after breeding as stated below, with expected ratios being no different from actual ratios of mice and included this data in Extended Data Fig. 1c:

“Nsun2 mutant mice were born and survived into adulthood at expected Mendelian ratios (Extended Data Fig. 1d) without overt health abnormalities and showed ~10% reductions in body and brain weight compared to WT littermates 14-16 weeks after birth (ANOVA; body weight, $p < 0.001$; brain weight, $p < 0.001$), with female but not male mutants exhibiting decreased brain/body ratio (t-test; $p < 0.01$; Extended Data Fig. 1d; Suppl. Table 1).”

6. *In Extended figure 2b, 5'tRF was also decreased, as well as the mature tRNA. Thus an instability-related mal-production of tRF can not explain the decrease of specific tRNAs. What is authors explanation about the distinct results in embryonic brain (Blanco S et al., 2014)?*

Response: We have now included a comprehensive introduction where we mention the canonical role of Nsun2 in tRNA fragmentation, citing the Blanco et al, 2014 study in embryonic brain in lines 71-74. We then discuss in lines 76-88 comprehensively how our work differs from previous studies including Blanco et al, as we detect a mechanism of translational control independent of tRNA fragmentation that is specific to the adult brain:

In the current study, we use three different genetic approaches selectively targeting NSUN2 function in differentiated neurons of the postnatal and adult brain. These include neuron-specific conditional Nsun2 ablations and transgene-mediated increase in Nsun2 expression and methylation activity in adult prefrontal cortex (PFC), thereby focusing for the first time on the effect of NSUN2 enzymatic activity on specific neuronal subpopulations in the mature brain. We report that Nsun2 shapes complex behaviors with neuronal function highly sensitive to bi-directional changes in tRNA methyltransferase activity. We show that the underlying mechanisms include alterations in tRNAs defined by high cytosine methylation specifically in the variable loop region, resulting in prominent deficits of glycine-specific tRNA isodecoders with corresponding shifts in the neuronal proteome due to decreased translational efficiency of glycine-rich neuronal proteins. Ultimately, these distortions in the glycinergic neuronal translome lead to a striking 2.46-fold increase in PFC glycine levels associated with multi-fold increases in several key enzymes of the glycine biosynthetic pathway, further illustrating that in the adult brain, proteomic and metabolic homeostasis associated with specific amino acids is linked to epitranscriptomic regulation of cognate tRNAs.

7. *No potential functional links between the proteome alteration or the e-phys results and behavior were provided or attempted.*

Response: We believe we addressed this comment in our response to comment #1 from this Reviewer as stated below:

This newly added text in the discussion starts with a lead-in sentence 'Which molecular mechanisms link Nsun2 to these robust behavioral and physiological phenotypes'? Lines 252-278 connect high Gly tRNA methylation to the fact that these tRNAs are selectively decreased after Nsun2 knock-out in mature neurons, then draw a link the decreased translational efficiencies of Gly-rich proteins including synaptic proteins, thereby drawing a link to the electrophysiological deficits and behavioral changes. To more directly address the Reviewer's comments, we now also included a new Table (Suppl. Table 7) and Figure (Extended Data Fig. 4f) listing 50 key synaptic proteins that are downregulated in mutant, for which we now show data for glycine composition. Extended Data Fig. 4f demonstrates that these 50 key synaptic proteins (downregulated in Nsun2 mutant) show an even higher mean glycine percentage.

In addition, we conducted for this resubmission additional metabolomic studies, showing a dramatic 146% increase in cortical glycine level, an alteration that was highly specific to this particular canonical amino acid. We now discuss the molecular alterations in the glycine biosynthesis pathway in our mutant mice in lines 279-304.

8. *NSUN2 targets a variety of tRNAs including glycine isodecoders. The quantitative results on other known NSUN2 target tRNAs need to be carefully investigated and described (doi: 10.3390/genes10020102).*

Response: We agree with the Reviewer that other Nsun2 targets are important to characterize in order to identify specificity for Gly tRNAs. We note that previous publications, such as Blanco et al (2014) have thoroughly characterized tRNA methylation across all isodecoders in the embryonic brain and therefore we did not conduct methylation assays on every tRNA. We originally included methylation data for Gly, Glu, and Asp because Gly and Glu are 2 out of 3 tRNAs that contain 3 Nsun2-mediated methylation sites, and Asp serves as a control for a known DNMT2 methylation site remaining unchanged after Nsun2 KO, as discussed in lines 252-278. We have now:

(i) added an additional experiment with methylation data for the third tRNA that contains 3 methylation sites (Pro) as well as another nuclear tRNA (ValAAC) to provide additional evidence for Nsun2 KO remaining consistent across Nsun2-targeted tRNAs (See Extended Data Fig. 3a).

(ii) created an additional two figure panels demonstrating the thorough data provided by Blanco and colleagues (2014) on number of Nsun2-methylation cytosines at each tRNA isoacceptor in mouse skin (Extended Data Fig. 3b) and compared this to our data from wild-type mouse cortex showing glycine as a tRNA that in brain has an additional Nsun2-methylation site (Extended Data Fig. 3c), which points to the specificity for Gly in our studies.

9. *In addition to tRNA, NSUN2 also targets vault RNA, an abundant lncRNA, and some mRNAs in neurons. The impact to these RNAs in the KO and OE mice should be described and explained (doi: 10.3390/genes10020102).*

Response: We agree that the effects of Nsun2 KO on all these species of RNA is an important future direction, and we have added in the Discussion lines 318-321 this information including references to studies that have previously characterized alterations to these RNAs after global Nsun2 KO.

10. *Authors claimed ketamine significant decrease in tRNAGlyGCC methylation 24 hours post-injection (Line 144-145), but no actual values were mentioned in the main text. The quantification graph in Fig 3g and data in Supplementary table 9 showed highly similar methylation level between saline and ketamine treated animals. If there is a reduction, the reduction is in a range of 0.0-0.3%. Is this level of reduction sufficient for linking the anti-depressant effect of ketamine and behavior in KO mice? This reviewer can not comfortably conclude the same with the authors.*

Response: We have removed the ketamine experiment from the manuscript due to the low magnitude of change pointed out by the Reviewer.

11. *Line 135-139. The conclusion on translational-control of glycine-rich proteins as the 3rd glycinergic regulatory pathway of central nervous system is not convincing.*

Response: As stated in response to Reviewer 3 comment 5 below, we believe uncovering a drastic change in glycine amino acid in the Nsun2 KO brain and changes in glycine biosynthetic proteins gives further cause to believe that Gly epitranscriptomic mechanisms are an additional glycinergic regulatory pathway in the mature brain:

(i) First, we noticed that within our proteomics data, there were 5 differentially expressed enzymes involved in glycine/serine biosynthesis, suggesting that glycine amino acid concentrations may be altered in the Nsun2 KO. We have now included an additional main Figure (Figure 3a) showing the glycine/serine biosynthesis pathways and using arrows to denote which enzymes were altered in the Nsun2 KO.

(ii) We then conducted an additional experiment with new cortical tissue samples in conjunction with the Rockefeller Proteomics Core, performing metabolomics for all amino acids to assess abundance in Nsun2 KO vs. WT. We did indeed find a dramatic increase in glycine in the Nsun2 KO, while other amino acids were unchanged, and we have added this data into the manuscript as stated below and in new Figure 3b. Further, we have added a paragraph about this change in amino acid biosynthesis into the Discussion section lines 279-304.

12. *There is no Discussion section.*

Response: We agree this manuscript would benefit greatly from a discussion section to interpret results and discuss limitations and future directions, and have now added a substantial discussion section in lines 246-323.

13. *To address the role of NSUN2 in PFC, a functional rescue using AAV8-Nsun2-GFP in KO background is preferred.*

Response: We thank the reviewer for making this point, which is essentially asking whether Nsun2 in the PFC is essential for complex behaviors. To address this we have:

(i) Added a third type of genetic experiment by localized ablation of Nsun2 in PFC specifically. We have conducted 2 additional validation experiments using this PFC-specific ablation of Nsun2 via AAV-Cre injection to produce a more direct link of causality for Nsun2 function in PFC, including qPCR for Nsun2 decrease after AAV-Cre injection and bisulfite amplicon sequencing of Gly tRNA showing decreased methylation. We have added this additional experiment into the text and into Extended Data Fig. 1d (validation of *Nsun2* expression and Gly tRNA methylation)

(ii) Performed tests of behavioral despair (TST and FST) on AAV-Cre injected mice and identified an anti-depressant phenotype that is presented in Fig. 5e and discuss these findings in lines 233-243.

We're not aware of an established approach for reversing gene expression deficits in a widespread forebrain KO model. Due to the many brain regions affected by the forebrain KO of Nsun2, it is likely there are other areas where Nsun2 function essential for certain behaviors, but we start here by isolating PFC and conducting the AAV-Cre-mediated KO and AAV-Nsun2 transgene overexpression in that region.

Minor points:

1. *Please provide primer sequences used for RT-PCR in this study.*

Response: We have now included primers used for RT-PCR in bisulfite amplicon sequencing of tRNAs as a new table, Suppl. Table 14 and refer to this in the methods section. For qPCR of *Nsun2*, we used Taqman probes and have listed the AssayID for *Nsun2* and *Gapdh* in the methods.

2. *What does the Y-axis in Fig 1a represent for the RNA-seq profile. At what age were the mice sampled?*

Response: We have now added the Y axis height on the RNAseq track for Nsun2 vs. KO in Figure 1a. Mice were sampled for RNAseq at ~8 weeks of age, which was added to the Methods section.

3. *In Fig 1b, no merged image is provided to show neuron-expression of NSUN2 in the cortex.*

Response: We agree with the reviewer that it is important to show Nsun2 staining in the cortex, but Figure 1b detected Nsun2-GFP from our overexpression virus. We have added a merged image to figure 1b to show Nsun2-GFP fusion protein staining merged with NeuN. Additionally, due to lack of good Nsun2 antibodies for fluorescence IHC in brain, we conducted a new experiment to confirm Nsun2 staining in cortex in WT and KO mice. We used RNA Basescope to perform *in situ hybridization* on cortical slices from layers II/III of wild type and Nsun2 KO cortex. We have included this data as a new panel in Extended Data Fig. 1b.

4. *The authors described Fig 1c as heatmaps, but they are not by definition. They are tables representing methylation status on each read. The number of rows should be described in the legend.*

Response: We understand the Reviewer's thought that tables in which we describe methylation aren't by definition heatmaps, so we have revised all text in the manuscript to call these "methylation maps"

instead of heatmaps. We have also added the number of rows (reads) to the y-axis for each methylation map in all methylation data-containing figures.

5. *There is no sufficient data suggesting Fig 3b.*

Response: We acknowledge that Fig. 3b (new Fig 4b) is not a concrete statement for the findings from our paper, but we now refer to this in the Figure legend and in the text as a “Working Model” that we think is important to convey to the reader the molecular findings and their relation to neurotransmission graphically.

6. *I could not find information to describe the ages of the mice used for each experiment.*

Response: We apologize for overlooking this, we used mice 10-12 weeks old for AAV injections and 13-16 weeks old for behavior testing and most molecular assays. We have added this information in the methods section as stated below:

“Mice were 13-16 weeks old for all behavioral experiments and sacrificed at 13-16 weeks for all molecular experiments except for tRNA bisulfite sequencing and RNA sequencing of *CamK-Cre Nsun2* KO mice, which were ~8 weeks old at time of sacrifice.”

7. *P value as 0.06 (Fig 3) was considered significant in this study. This is not conventional, please provide rationale for significant.*

Response: We have removed the wording that this was significant as we realize this is not a conventionally significant result.

8. *In Fig 2a and Extended Fig 2a, statistical analysis results are not depicted.*

Response: Figure 2a statistical results are listed in Suppl. Table 2, and Extended Fig. 2a (now Extended Fig. 4a) are listed in Suppl. Table 4. To ensure the reader will not overlook these data, we have added text when we refer to these tables next to the figure reference as shown below:

“Fig. 2a, statistical analyses for all detected isodecoders in Suppl. Table 2)” and “(Extended Data Fig. 4a; statistical analyses for all detected isodecoders in Suppl. Table 4)

Reviewer #3:

In their manuscript Blaze and colleagues demonstrate that NSUN2-mediated tRNA methylation is required for the efficient translation of glycine-rich mRNAs in the postnatal mouse brain. The codon-specific shift of translation affects the global neural proteome and alters synaptic signalling and behaviour.

tRNA is the most extensively modified RNA, and many of these modifications have been linked to human diseases including neurological disorders. Mutations in the tRNA methyltransferase NSUN2 has been widely linked to neurological deficits in mouse and human. However, all previous studies focused on the role of NSUN2-specific tRNA methylation during development. Hence, the functional importance of NSUN2 in the postnatal brain is currently unknown. Moreover, the precise underlying molecular mechanisms of how NSUN2-mediated methylation controls mRNA translation rates are unclear. Here, the authors show for the first time that NSUN2-specific methylation of tRNAs determines codon usage. Given that dysfunctional RNA processing and translation in neuronal tissue is often observed in neurological diseases, this study is a timely and important contribution to the field. Overall, this is a very mature manuscript but is very dense, which makes it at times difficult to follow and the novel findings are at times a little lost.

1. Page 2; line 49: Only when reading the methods section, it becomes clear that the generated lines deplete NSUN2 postnatally using the CamK-Cre mice. This is very different from published mouse lines where only the developing brain was analysed. The function of NSUN2 in the postnatal brain is entirely unknown. Moreover, over-expression experiments have also not been performed before. I recommend highlighting the two facts better and early on in the text.

Response: This comment is similar to a comment made by Reviewer 1, and we point to our response (listed below):

(i) added a comprehensive introductory paragraph (Line 66-76) describing the previous work on Nsun2-induced tRNA methylation:

Evidence for the importance of solely NSUN2-mediated m⁵C in human health and disease was first provided by individuals with loss-of-function mutation in NSUN2, displaying intellectual disability (ID), facial dysmorphism and distal myopathy¹⁰⁻¹³. The idea that NSUN2 deficiency causes neurological abnormalities was further delineated by studies in Drosophila (d) and mouse^{9,10}, including dNsun2 knockdown with impaired short-term memory after aversive olfactory conditioning that was rescued by pan-neuronal expression of dNsun2¹⁰. Furthermore, mice with Nsun2 germline deletion demonstrated various impairments in locomotor activity and behavior together with reduced brain size due to excessive cell death in prenatal brain⁹. The mechanism thus far suggested for these deficits is impaired translation induced by increased tRNA fragmentation after Nsun2 ablation^{14,15}. However, although tRNA cytosine methylation (m⁵C) has recently been profiled in various tissues including embryonic brain^{9,16-22} the m⁵C RNA methylome and its functional relevance in the mature adult mammalian brain is still unexplored.

(ii) We then added (Line 76-88) added a comprehensive paragraph describing how our work differs from previous studies to show new functions of Nsun2 in adult brain function:

In the current study, we use three different genetic approaches selectively targeting NSUN2 function in differentiated neurons of the postnatal and adult brain. These include neuron-specific conditional Nsun2 ablations and transgene-mediated increase in Nsun2 expression and methylation activity in adult prefrontal cortex (PFC), thereby focusing for the first time on the effect of NSUN2 enzymatic activity on specific neuronal subpopulations in the mature brain. We report that Nsun2 shapes complex behaviors with neuronal function highly sensitive to bi-directional changes in tRNA methyltransferase activity. We show that the underlying mechanisms include alterations in tRNAs defined by high cytosine methylation specifically in the variable loop region, resulting in prominent deficits of glycine-specific tRNA isodecoders with corresponding shifts in the neuronal proteome due to decreased translational efficiency of glycine-rich neuronal proteins. Ultimately, these distortions in the glycinergic neuronal translatoome lead to a striking 2.46-fold increase in PFC glycine levels associated with multi-fold increases in several key enzymes of the glycine biosynthetic pathway, further illustrating that in the adult brain, proteomic and metabolic homeostasis associated with specific amino acids is linked to epitranscriptomic regulation of cognate tRNAs.

2. Page 3; line 83: Can the authors comment on why glycine is specifically affected? Could it be that this is simply due to the fact that in the brain all 3 cytosines in the variable loop are methylated by NSUN2. So, tRNAGly would be hypermethylated when compared to other tissues?

Response: We appreciate the Reviewer's suggestion and agree that likely the specificity of tRNAGly is due to the four Nsun2-methylated cytosines at and around the variable loop, whereas the other tRNAs with three Nsun2-methylated cytosines are not as highly methylated. As discussed in comment 10 from Reviewer 2:

We note that previous publications, such as Blanco et al (2014) have thoroughly characterized tRNA methylation across all isodecoders in the embryonic brain and therefore we did not conduct methylation assays on every tRNA. We originally included methylation data for Gly, Glu, and Asp because Gly and Glu are 2 out of 3 tRNAs that contain 3 Nsun2-mediated methylation sites, and Asp serves as a control for a known DNMT2 methylation site remaining unchanged after Nsun2 KO, as discussed in lines 252-278. We have now:

(i) added an additional experiment with methylation data for the third tRNA that contains 3 methylation sites (Pro) as well as another nuclear tRNA (ValAAC) to provide additional evidence for Nsun2 KO remaining consistent across Nsun2-targeted tRNAs (See Extended Data Fig. 3a).

(ii) created an additional two figure panels demonstrating the thorough data provided by Blanco and colleagues (2014) on number of Nsun2-methylation cytosines at each tRNA isoacceptor in mouse skin (Extended Data Fig. 3b) and compared this to our data from wild-type mouse cortex showing glycine as a tRNA that in brain has an additional Nsun2-methylation site (Extended Data Fig. 3c), which points to the specificity for Gly in our studies.

3. *Page 4; line 92: Are neural proteins in general enriched in Gly when compared to all proteins?*

Response: This comment is similar to a comment made by Reviewer 2, and we point to our response to that comment below:

We agree with the Reviewer that because our Nsun2 KO was specific to Camk2a-expressing excitatory glutamatergic neurons, it is likely that the downregulated proteins are in that cell type, and we have now noted in the discussion that a limitation of our proteomic experiments is the lack of cell specificity due to input restrictions. We have now added an additional analysis to our proteomics data to attempt to answer this question. To identify whether the repression of glycine-rich proteins was due to the neuronal proteome, and further glutamatergic neuron proteins, being enriched for glycine, we conducted additional analyses and have specified in the Results section lines 151-156 the following text as well as adding the new data to new Extended Data Fig. 4f:

“These alterations were highly specific because glycine content of the entire set of n=434 neuron-specific proteins (UniProtKB) in our cortical proteome (comprised of neuronal and glial proteins) was very similar to the glycine content of the total proteome in our cortex homogenates (Mann-Whitney test, p=0.412; Extended Data Fig. 4f), confirming that the observed enrichment of high glycine content proteins in the fraction of down-regulated proteins in the Nsun2 mutant cortex do not reflect a non-specific decline of the neuronal proteome overall.”

4. *I believe figure 3b is not referred to in the text.*

Response: We have added a direct reference to Figure 3b (now Figure 4b) in the text.

5. *If glycine-rich mRNAs are less translated, does this cause an increase of the amino acid glycine in the cells?*

Response: We agree that this was a crucial question to answer for this manuscript, so we used a two-pronged approach to answer this question:

(i) First, we noticed that within our proteomics data, there were 5 differentially expressed enzymes involved in glycine/serine biosynthesis, suggesting that glycine amino acid concentrations may be altered in the Nsun2 KO. We have now included an additional main Figure (Figure 3a) showing the glycine/serine biosynthesis pathways and using arrows to denote which enzymes were altered in the Nsun2 KO.

(ii) We then conducted an additional experiment with new cortical tissue samples in conjunction with the Rockefeller Proteomics Core, performing metabolomics for all amino acids to assess abundance in Nsun2 KO vs. WT. We did indeed find a dramatic increase in glycine in the Nsun2 KO, while other amino acids were unchanged, and we have added this data into the manuscript as stated below and in new Figure 3b. Further, we have added a paragraph about this change in amino acid biosynthesis into the Discussion section lines 279-304.

Reviewers' Comments:

Reviewer #1:

Remarks to the Author:

In general, we find the manuscript improved. The addition of the ribo-seq experiment provides additional confidence in the results.

We nonetheless have two remaining points:

In our previous first point in the review we asked the authors to better define the boundaries between their novel findings and what's known. As we wrote, "The fact that tRNA-Gly is a target of nsun2 is well-established. Also the fact that tRNA-Gly stability is nsun2 dependent has been characterized by studies by the Suzuki and Frye groups". The authors have added significant text in the introduction about what's known about m5C and associated phenotypes in the brain, yet they make no reference to the fact that tRNA-Gly was previously identified as a key tRNA target of nsun2 by the Suzuki and Frye groups. This needs to be explicitly mentioned in the introduction, and should not be presented as a novel finding on behalf of this manuscript.

In addition, while we believe the ribo-seq is an important result, unfortunately the current presentation of the results in Fig. 2e is completely unclear – what do the colors represent? What is the Y axis? Also, why is data shown only for a subset of codons and not for all of them? None of this is clarified in the legends. Can the authors present instead a simple graph showing the mean estimated dwell time as a Y axis and each of the codons on the X axis?

Reviewer #2:

Remarks to the Author:

This reviewer appreciates the authors' rigorous revision. All concerns raised in my previous report have been addressed to satisfaction. I only have two minor concerns /comments as following:

1. Line 38-39, the authors state that epitranscriptomic mechanisms linking the brain proteome to cognition and complex behaviors essentially remain unexplored. In my opinion, this is not true. Shi et al 2018 in Nature has published the role of m6A reader YTHDF1 in learning and memory; Engel et al 2018 in Neuron has published the m6A regulational response to stress; Widagdo et al 2016, Walters et al 2017 published m6A regulation in fear memory; in addition to reports in other organisms.

2. Line 198-199, "remaining 17 amino acids", 18?

Reviewer #3:

Remarks to the Author:

The authors have addressed all my initial concerns either by providing more data or through clarifying the text.

Reviewer #4:

Remarks to the Author:

The presented work uses a multidisciplinary approach to elucidate the role of the tRNA cytosine methyltransferase Nsun2 in the brain, both on the molecular as well as at the physiological level. Understanding how epitranscriptomic modifications translate into biology is a very challenging task and an important topic to be addressed. The work is based on a tremendous amount of data combining almost all relevant disciplines. I appreciate the authors' efforts to further back-up results by independent approaches, to employ several controls and to use rather unbiased -omics experiments to draw their conclusions. Given the complexity of the problem and the limitations in the amount of work that can be reasonably fit in a single paper, the study inevitably falls short when it comes to linking the molecular findings to the observed phenotype. In my view, however,

this should not prevent publication of this work provided that some adjustments are made:

1. Throughout the ms the authors use the term „regulation“ to describe any changes they observe upon manipulation of Nsun2. As long as no clear evidence is provided that Nsun2 is part of a biological regulatory process (i.e. an adaptive change responding to a specific developmental, physiological or biological need), the use of „regulation“ should be strictly avoided.
2. Abstract line 45/46: this is an overstatement since no causative mechanism could be concluded neither should it be suggested based on the available data (see also my comments below).
3. Results, line 94: The Western blot signals in Fig. 1b/S1c suggest that levels of Nsun2 protein were only mildly affected by neuronal ko (far from being ablated). Unfortunately, no further details or explanations are given. Does this reflect stable Nsun2 protein remaining, expression of Nsun2 in other cell types, incomplete ko or a mixture of these effects? How do the authors ensure effective ko of the protein in all forebrain neurons?
4. Fig. 1a/b / line 93/94: Please correct the labeling / referral to the subfigures. Fig. 1b shows local overexpression, not neuron-specific ablation.
5. Fig. 1c/d / line 112: Labeling in the subfigures and text don't match (tRNA^{Asp}GTC C38 vs C37).
6. Fig. 2a: The effect on Gly tRNA isodecoders is indeed striking. Is there a mismatch between the two panels? I count 6 green dots about 50% reduced (left) and 7 Gly-tRNA isodecoder bars reduced to 50% (right)?
7. Fig. 2a: Given the rather symmetric distribution of fold-changes, which tRNAs were strongly increased (besides the examples shown)? Did the authors also check their potential impact on protein translation efficiency?
8. line 61, „meta analysis“
9. line 141: the use of the terms „up-“ and „down-regulated“ is particularly inadequate in this context.
10. Fig. 2b/c: The high number of protein abundance changes may on one hand be not so surprising. The potential implications of these changes, on the other hand, are less clear from the presentation of solely statistical values. At least a plot should be added depicting each protein by its average abundance ratio KO/WT and the standard deviation of this ratio (this may be done by random pairing or, better, permuted pairing of datasets).
11. Fig. 2c, left: The pathway annotations are suggestive but should be interpreted with more care. For example, proteomic analyses are typically biased towards higher-abundant proteins (which are not evenly distributed over functional categories), categories are not all defined with the same stringency and knowledge base, or classification of proteins with multiple functions can be quite arbitrary.
12. Fig. 2c, right: It would be more meaningful to analyze whether there is a clear correlation between the observed (average) abundance change of a protein with its relative content of affected (apparently three of the four glycine codons used in mice) rather than total glycine content (which seems not really discriminative). In case the original hypothesis holds, this would also help to better differentiate between proteins directly affected by Nsun2 ko and proteins that may have changed due to secondary effects.
13. Lines 149ff, 159ff: This is an over-interpretation of the data and should be omitted. The number of proteins in that class is high, differences in glycine content are marginal, therefore selected examples do not make the point.
14. Fig. 2d: Again, why did the authors use total Gly codon content rather than the affected subset?
15. Fig. 2d, right: Pearson correlation is missing in my version of the ms. The numbers given in the legend suggest that (anti)correlation coefficients are quite indiscriminative. Please explain.
16. Reading the data in Fig. 2, I am getting more and more confused about the codon specificity of the Nsun2 ko effects. 2a lists CCC, GCC and TCC (but not ACC) tRNA isodecoders as significantly decreased. Then, 2d describes GCC and CCC as anti-correlated with logTER but TCC and ACC not associated. And 2e shows increase of ribosome dwell time only for GGA/TCC, but not for any of the other Gly-specific codons. These discrepancies should be clarified in the ms.
17. Fig. 3a: Correct labeling „Serine racemase“.
18. Fig. 3a: In addition of p-value changes, average fold changes should be indicated.
19. Fig. 4 / lines 205-219: This is the weakest set of data in the manuscript. The conducted experiments are too preliminary, if not to say naive, and the results inconclusive at this stage. For example, pre- and postsynaptic phenotypes would imply very different pools of proteins, and the small differences observed can not be attributed to changes of specific (or the ensemble of)

synaptic proteins as suggested. More sophisticated electrophysiological experiments combined with targeted manipulation of individual proteins and controls would be required to establish mechanistic link(s) to the observed behavioral phenotype(s), clearly reaching far beyond this manuscript. I strongly suggest to leave this part out of the manuscript. There is no need to speculate or to build hypotheses linking the complex molecular findings to the equally complex phenotype.

20. Fig. 5: The finding that Nsun2 ko has effects in behavioral tests does not mean that Nsun2 acts as a modulator under native conditions, please rephrase.

Dear Reviewers,

We very much appreciate the highly constructive comments of each of the four Reviewers and provide below a detailed point-by-point response. We are pleased to report that we were able to address each of the Reviewers' comments.

Reviewer #1

In general, we find the manuscript improved. The addition of the ribo-seq experiment provides additional confidence in the results. We nonetheless have two remaining points:

1. In our previous first point in the review we asked the authors to better define the boundaries between their novel findings and what is known. As we wrote, The fact that tRNA-Gly is a target of nsun2 is well-established. Also the fact that tRNA-Gly stability is nsun2 dependent has been characterized by studies by the Suzuki and Frye groups. The authors have added significant text in the introduction about what is known about m5C and associated phenotypes in the brain, yet they make no reference to the fact that tRNA-Gly was previously identified as a key tRNA target of nsun2 by the Suzuki and Frye groups. This needs to be explicitly mentioned in the introduction, and should not be presented as a novel finding on behalf of this manuscript.

Response: We refer to these previous and important findings in the Discussion section in lines 288-289 as follows:

“Of note, deficits in Gly tRNA levels after germline *Nsun2* ablation have been previously reported for liver^{33,54} and skin¹⁰.”

We now have also added text in the Introduction in lines 75-83 to emphasize these previous findings before we discuss our own study:

“While there is evidence for global impairment of protein synthesis via fluorescent labelling of nascent proteins in embryonic mouse brain¹⁵ and adult mouse skin¹⁷, to our knowledge no one has used unbiased proteomic approaches such as mass spectrometry to directly measure specific molecular pathways altered by *Nsun2* ablation in any tissue type. It has been established previously that global *Nsun2* knockout in mouse alters the pool of mature tRNAs in embryonic brain¹⁵ and adult skin¹⁵ and liver¹⁸, including a marked depletion of Gly tRNAs, which are a key target of *Nsun2* methylation. However, although tRNA cytosine methylation (m5C) has recently been profiled in various tissues including embryonic brain^{10,19-25}, the molecular characterization of the *Nsun2*-dependent epitranscriptome and its functional relevance to in the mature adult mammalian brain is still completely unexplored.

2. In addition, while we believe the ribo-seq is an important result, unfortunately the current presentation of the results in Fig. 2e is completely unclear what do the colors represent? What is the Y axis? Also, why is data shown only for a subset of codons and not for all of them? None of this is clarified in the legends. Can the authors present instead a simple graph showing the mean estimated dwell time as a Y axis and each of the codons on the X axis?

Response: We apologize for the lack of clarity in our Riboseq figure and appreciate the Reviewer's suggestion to show an XY plot with all codons. We have eliminated this confusing figure and made multiple new figure panels (Fig. 2e,f; Extended Data 5e) with

a schematic of ribosome A, P, and E site location and with multiple plots and additional analyses of dwell time and A-site occupancy grouped by amino acid and by codon. We describe this data and analyses in lines 191-205 in the revised manuscript.

Reviewer #2

This reviewer appreciates the authors rigorous revision. All concerns raised in my previous report have been addressed to satisfaction. I only have two minor concerns /comments as following:

1. Line 38-39, the authors state that epitranscriptomic mechanisms linking the brain proteome to cognition and complex behaviors essentially remain unexplored. In my opinion, this is not true. Shi et al 2018 in Nature has published the role of m6A reader YTHDF1 in learning and memory; Engel et al 2018 in Neuron has published the m6A regulational response to stress; Widagdo et al 2016, Walters et al 2017 published m6A regulation in fear memory; in addition to reports in other organisms.

Response: In the abstract, we clarify that we are referring to tRNA epitranscriptomic changes as follows:

“Epitranscriptomic mechanisms linking tRNA function and the brain proteome to cognition and complex behaviors essentially remain unexplored.”

We also have added the suggested references in the discussion to emphasize the important work that has been done linking another epitranscriptomic mark to brain function and behavior in lines 352-358.

2. Line 198-199, remaining 17 amino acids , 18?

Response: We thank the Reviewer for noticing this typo and have changed this to 18.

Reviewer #3

The authors have addressed all my initial concerns either by providing more data or through clarifying the text.

Reviewer #4

The presented work uses a multidisciplinary approach to elucidate the role of the tRNA cytosine methyltransferase Nsun2 in the brain, both on the molecular as well as at the physiological level. Understanding how epitranscriptomic modifications translate into biology is a very challenging task and an important topic to be addressed. The work is based on a tremendous amount of data combining almost all relevant disciplines. I appreciate the authors efforts to further back-up results by independent approaches, to employ several controls and to use rather unbiased omics experiments to draw their conclusions. Given the complexity of the problem and the limitations in the amount of work that can be reasonably fit in a single paper, the study inevitably falls short when it comes to linking the molecular findings to the observed phenotype. In my view, however, this should not prevent publication of this work provided that some adjustments are

made:

1. Throughout the ms the authors use the term *regulation* to describe any changes they observe upon manipulation of *Nsun2*. As long as no clear evidence is provided that *Nsun2* is part of a biological regulatory process (i.e. an adaptive change responding to a specific developmental, physiological or biological need), the use of *regulation* should be strictly avoided.

Response: We noticed that we often refer to epitranscriptomic regulation and have therefore changed these instances to epitranscriptomic modification.

2. Abstract line 45/46: *this is an overstatement since no causative mechanism could be concluded neither should it be suggested based on the available data (see also my comments below).*

Response: To avoid overstating these findings, we have now changed to wording to: "Loss of *Gly*-rich proteins critical for glutamatergic neurotransmission was associated with impaired synaptic signaling..."

3. Results, line 94: *The Western blot signals in Fig. 1b/S1c suggest that levels of Nsun2 protein were only mildly affected by neuronal ko (far from being ablated). Unfortunately, no further details or explanations are given. Does this reflect stable Nsun2 protein remaining, expression of Nsun2 in other cell types, incomplete ko or a mixture of these effects? How do the authors ensure effective ko of the protein in all forebrain neurons?*

Response: We have added information in the main text regarding the tissue used for Western blot and qPCR assays to measure *Nsun2* protein and gene expression after KO. Due to lack of cell type specificity in these assays and the fact that 30% of cortical neurons are *Camk2a* positive, it makes sense that our western blot results show ~30% decrease in *Nsun2*. We have added this information in the text lines 104-108.

4. Fig. 1a/b / line 93/94: *Please correct the labeling / referral to the subfigures. Fig. 1b shows local overexpression, not neuron-specific ablation.*

Response: We thank the Reviewer for noticing this oversight and we have corrected this in the text.

5. Fig. 1c/d / line 112: *Labeling in the subfigures and text don t match (tRNAAspGTC C38 vs C37).*

Response: We thank the Reviewer for noticing this oversight and we have corrected Fig. 1d to represent tRNAAspGTC C38.

6. Fig. 2a: *The effect on Gly tRNA isodecoders is indeed striking. Is there a mismatch between the two panels? I count 6 green dots about 50% reduced (left) and 7 Gly-tRNA isodecoder bars reduced to 50% (right)?*

Response: The 7 Gly-isodecoder bars on the right panel in Fig. 2a represent all the significantly altered (adjusted $p < 0.05$), while the dots on the left-hand figure represent all Gly isodecoders regardless of significance. We have added black circles around the

isodecoders that are significantly altered on the left-hand graph and have noted this in the figure legend.

7. Fig. 2a: Given the rather symmetric distribution of fold-changes, which tRNAs were strongly increased (besides the examples shown)? Did the authors also check their potential impact on protein translation efficiency?

Response: According to HydrotRNAseq (shown in Fig. 2a, right panel), we only found two tRNAs significantly increased in the KO cortex. We have confirmed with our codon-specific Riboseq data that codons associated with these two tRNAs do not show changes in ribosome dwell time and have added this to the Results section as follows:

“We further confirmed that codons associated with the two tRNAs that were upregulated in the Nsun2 KO cortex (see Fig. 2a; tRNA^{Asn}_{GTT} and tRNA^{Met}_{CAT}) did not have any significant changes in ribosome DT or A-site occupancy (Fig. 2f, Extended Data 5e).”

8. line 61, meta analysis

Response: We have fixed this error in the manuscript text.

9. line 141: the use of the terms up- and down-regulated is particularly inadequate in this context.

Response: We have changed the wording to “increased” and “decreased” in this context.

10. Fig. 2b/c: The high number of protein abundance changes may on one hand be not so surprising. The potential implications of these changes, on the other hand, are less clear from the presentation of solely statistical values. At least a plot should be added depicting each protein by its average abundance ratio KO/WT and the standard deviation of this ratio (this may be done by random pairing or, better, permuted pairing of datasets).

Response: We have added a plot as Extended Data Fig. 4g showing the KO/WT ratio (T-test difference) of protein abundance for each protein vs. standard deviation. We used mass spectrometry protein expression data from Fig. 2b with KO vs. WT differences plotted against standard deviations for each protein. The standard deviation was calculated by selecting three random, but non-repeating, replicates from each group and taking the average from them. Then a simple standard deviation was calculated between the two averages to provide a KO vs. WT ratio plotted against a standard deviation in a randomized matter. Proteins of interest all lie within the dashed red boxes with low standard deviations.

11. Fig. 2c, left: The pathway annotations are suggestive but should be interpreted with more care. For example, proteomic analyses are typically biased towards higher-abundant proteins (which are not evenly distributed over functional categories), categories are not all defined with the same stringency and knowledge base, or classification of proteins with multiple functions can be quite arbitrary.

Response: We acknowledge that Gene Ontology and pathway annotation can be somewhat arbitrary and that the nature of a proteomics is stochastic and tends to identify

the most abundant proteins. However, we have been minimizing these issues by performing a very deep fractionation to maximize the number of quantified proteins. The final quantitation is based on t-test statistics and fold-change, and not protein abundance. The annotation analyses have been FDR -corrected and used to further form a hypothesis that was tested using an orthogonal technique. Based on these arguments we believe we have interpreted the gene ontology analyses to an appropriate extent.

12. Fig. 2c, right: It would be more meaningful to analyze whether there is a clear correlation between the observed (average) abundance change of a protein with its relative content of affected (apparently three of the four glycine codons used in mice) rather than total glycine content (which seems not really discriminative). In case the original hypothesis holds, this would also help to better differentiate between proteins directly affected by Nsun2 ko and proteins that may have changed due to secondary effects.

Response: We agree with the Reviewer that it is important to differentiate between affected codons and total glycine. We used this figure as a preliminary step by measuring amino acid content in proteins down- and up-regulated after Nsun2, followed by the sensitive analysis of the FIRE heatmap for all glycine codons (Fig. 2d), and then finally used codon-specific analysis in ribosome profiling for the most specific method (see new text lines 190-205 and new Fig. 2e-f).

13. Lines 149ff, 159ff: This is an over-interpretation of the data and should be omitted. The number of proteins in that class is high, differences in glycine content are marginal, therefore selected examples do not make the point.

Response: This additional analysis was added upon the request of a previous Reviewer in the initial review as shown below:

Reviewer comment: *Is it possible that the downregulation in the proteome was due to overall repression of proteins in excitatory neurons instead of restricted repression of glycine-rich proteins? Since the proteome analysis is done on tissue lysate of a mixture of all cell types including neurons, functional categories related to excitatory neuron function such as glutamatergic synapse can become enriched when the repression occurs in KO neurons.*

Response: We agree with the Reviewer that because our Nsun2 KO was specific to Camk2a-expressing excitatory glutamatergic neurons, it is likely that the downregulated proteins are in that cell type, and we have now noted in the discussion that a limitation of our proteomic experiments is the lack of cell specificity due to input restrictions. We have now added an additional analysis to our proteomics data to attempt to answer this question. To identify whether the repression of glycine-rich proteins was due to the neuronal proteome, and further glutamatergic neuron proteins, being enriched for glycine, we conducted additional analyses and have specified in the Results section lines 164-169 the following text as well as adding the new data to new Extended Data Fig. 4f:

“These alterations were highly specific because glycine content of the entire set of n=434 neuron-specific proteins (UniProtKB) in our cortical proteome (comprised of neuronal and glial proteins) was very similar to the glycine content of the total proteome

in our cortex homogenates (Mann-Whitney test, $p=0.412$; Extended Data Fig. 4f), confirming that the observed enrichment of high glycine content proteins in the fraction of down-regulated proteins in the Nsun2 mutant cortex do not reflect a non-specific decline of the neuronal proteome overall. ”

14. *Fig. 2d: Again, why did the authors use total Gly codon content rather than the affected subset?*

Response: We have clarified in the manuscript that we computed translation efficiencies and compared to all Gly codons combined because of the decrease in all four isoacceptor families in our main tRNA seq dataset. We have added to the manuscript line 173: “...including all four Gly codons due to the unanimous decrease in the Nsun2 KO”

15. *Fig. 2d, right: Pearson correlation is missing in my version of the ms. The numbers given in the legend suggest that (anti)correlation coefficients are quite indiscriminative. Please explain.*

Response: We apologize for the oversight in this figure legend. We removed these regression-based comparisons between codon content and logTER as we find them less reliable than the codon-specific Riboseq and the FIRE heatmaps. We did not intend to include Pearson correlations for specific anticodons in the final manuscript and have omitted this data and now have omitted this from the Fig. 2 legend.

16. *Reading the data in Fig. 2, I am getting more and more confused about the codon specificity of the Nsun2 ko effects. 2a lists CCC, GCC and TCC (but not ACC) tRNA isodecoders as significantly decreased. Then, 2d describes GCC and CCC as anti-correlated with logTER but TCC and ACC not associated. And 2e shows increase of ribosome dwell time only for GGA/TCC, but not for any of the other Gly-specific codons. These discrepancies should be clarified in the ms.*

Response: We apologize for the lack of clarity about codon-specificity in the manuscript. In Fig. 2a (left), we show that all tRNA isodecoders for Gly are decreased in the Nsun2 KO, but only some (including isodecoders for the anticodons CCC, GCC, and TCC) are statistically significant (Fig. 2a right). As stated in the previous comment, we did not intend to include Pearson correlations for specific anticodons in the final manuscript. We find this analysis to be less rigorous and reliable than the FIRE heatmaps and RiboSeq analyses so we had omitted this data and now have omitted this from the Fig. 2 legend.

17. *Fig. 3a: Correct labeling Serine racemase .*

Response: We have corrected this typo in Fig. 3a.

18. *Fig. 3a: In addition of p-value changes, average fold changes should be indicated.*

Response: We have added average t-test differences next to each arrow where there is a significant increase or decrease in Fig. 3a.

19. *Fig. 4 / lines 205-219: This is the weakest set of data in the manuscript. The conducted experiments are too preliminary, if not to say naive, and the results inconclusive at this stage. For example, pre- and postsynaptic phenotypes would imply very different pools of proteins, and the small differences observed can not be attributed to changes of specific (or the ensemble of) synaptic proteins as suggested. More sophisticated electrophysiological experiments combined with targeted manipulation of individual proteins and controls would be required to establish mechanistic link(s) to the observed behavioral phenotype(s), clearly reaching far beyond this manuscript. I strongly suggest to leave this part out of the manuscript. There is no need to speculate or to build hypotheses linking the complex molecular findings to the equally complex phenotype.*

Response: We agree with the Reviewer's comment that this is a preliminary finding and there is much additional work to be able to establish a thorough mechanistic link between proteins, synaptic transmission phenotypes, and behavior, and we have added to line 279 of the text that the electrophysiological data is preliminary in nature. However, we think based on the other 3 reviewers comments and editor's request that it is best to leave this component in the manuscript to show an important phenotypic consequence of *Nsun2* ablation. We have also added an additional experiment showing decreased expression of Neurogranin, the synaptic protein with highest glycine content, in *Nsun2* KO cortical synaptosomes specifically, making a clearer link between synaptic physiology and glycine-rich protein abundance (Fig 4b).

20. *Fig. 5: The finding that *Nsun2* ko has effects in behavioral tests does not mean that *Nsun2* acts as a modulator under native conditions, please rephrase.*

Response: We have changed the wording in this figure to reflect that "manipulation of *Nsun2*" modulates complex behaviors so as not to imply *Nsun2* modulates these behaviors under native conditions.

Reviewers' Comments:

Reviewer #4:

Remarks to the Author:

The authors have appropriately addressed all my concerns raised.
I therefore can recommend publication of the manuscript.